# Periodic Graph Transformers for Crystal Material Property Prediction

**Keqiang Yan**
Computer Science & Engineering
Texas A&M University
College Station, TX 77843
keqiangyan@tamu.edu

**Yi Liu**[*]
Computer Science
Florida State University
Tallahassee, FL 32306
liuy@cs.fsu.edu

**Yuchao Lin**
Computer Science & Engineering
Texas A&M University
College Station, TX 77843
kruskallin@tamu.edu

**Shuiwang Ji**[*]
Computer Science & Engineering
Texas A&M University
College Station, TX 77843
sji@tamu.edu

## Abstract

We consider representation learning on periodic graphs encoding crystal materials. Different from regular graphs, periodic graphs consist of a minimum unit cell repeating itself on a regular lattice in 3D space. How to effectively encode these periodic structures poses unique challenges not present in regular graph representation learning. In addition to being E(3) invariant, periodic graph representations need to be periodic invariant. That is, the learned representations should be invariant to shifts of cell boundaries as they are artificially imposed. Furthermore, the periodic repeating patterns need to be captured explicitly as lattices of different sizes and orientations may correspond to different materials. In this work, we propose a transformer architecture, known as Matformer, for periodic graph representation learning. Our Matformer is designed to be invariant to periodicity and can capture repeating patterns explicitly. In particular, Matformer encodes periodic patterns by efficient use of geometric distances between the same atoms in neighboring cells. Experimental results on multiple common benchmark datasets show that our Matformer outperforms baseline methods consistently. In addition, our results demonstrate the importance of periodic invariance and explicit repeating pattern encoding for crystal representation learning. Our code is publicly available at https://github.com/YKQ98/Matformer.

## 1 Introduction

Crystal material property prediction is important for the discovery of new materials with desirable properties [38, 35–37, 50, 52, 41, 4, 33, 6]. Different from molecules and proteins [15, 51, 42, 9, 49, 26, 44], which are commonly represented as graphs [45, 29, 11, 12, 25, 27, 34], crystals consist of a minimum unit cell repeating itself on a regular lattice in 3D space. Thus, crystals are naturally represented as periodic graphs. A key challenge of crystal material property prediction lies in how to effectively encode periodic structures that are not present in regular molecular graph representations [10, 20, 41, 23, 22, 31, 24, 39, 17, 13, 1, 47]. E(3) invariance for molecular graphs requires the representation for a given molecule to be invariant to translation, rotation and reflection

---

[*]Equal senior contributions

36th Conference on Neural Information Processing Systems (NeurIPS 2022).

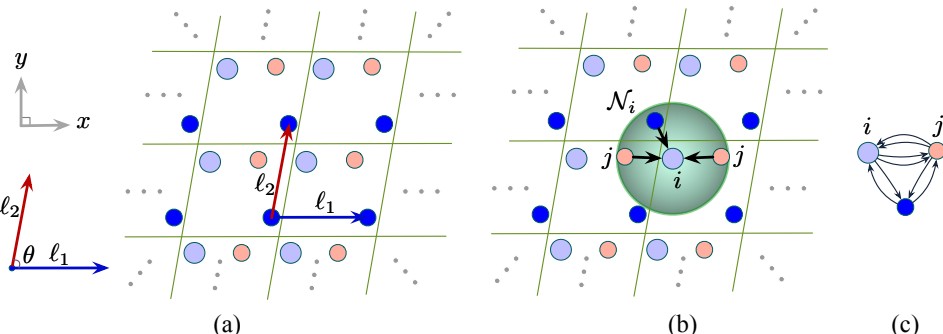

Figure 1: Illustrations of periodic patterns and the multi-edge graph construction method. Green lines are artificial boundaries to form one possible unit cell that repeats in infinite space for the given crystal. (a). An illustration of periodic patterns in 2D space. We use blue and red arrows to show how the blue atom repeats itself along $\ell_1$ and $\ell_2$. We show a general case in nature that $\ell_1$ and $\ell_2$ are not orthogonal. In this specific example, $\theta$ is less than $\frac{\pi}{2}$. (b) and (c). Illustrations of the multi-edge graph construction method and the constructed graph. The green circle shows atom $i$'s neighborhood $\mathcal{N}_i$, and black arrows are edges from atom $i$'s neighbors to itself. As atom $j$ repeats twice within $i$'s neighborhood in this example, the multi-edge graph construction method builds two edges from atom $j$ to atom $i$. The crystals are 3D structures in practice, and we use illustrations in 2D for simplicity.

transformations in 3D space. Beyond that, periodic graphs of crystals require unique periodic invariance. Periodic invariance has two facets; those are, the learned representations should be invariant to both the scaling up of the minimum repeatable unit cells and shifts of periodic boundaries. Although the former has been considered in Xie et al. [52, 53], the later is rarely identified explicitly and sometimes not considered by previous studies [52, 4, 46, 41, 6, 53, 33, 54, 1, 5]. Furthermore, the repeating patterns of periodic graphs should be captured explicitly. Given a fixed minimum unit cell structure, lattices of different sizes and orientations may correspond to different materials. Without such periodic patterns, the infinite structures of crystals may not be represented accurately. However, the explicit encoding of periodic patterns is not explored in previous studies [52, 4, 41, 6, 33, 1].

In this work, we propose to tackle periodic graph representation learning by incorporating both periodic invariance and periodic pattern encoding. We propose a periodic graph transformer, known as Matformer, that is periodic invariant and can capture periodic repeating patterns explicitly for crystal representation learning. Matformer achieves periodic invariance through two uniquely designed graph construction methods. It further encodes periodic patterns by efficient use of geometric distances between the same atoms in neighboring cells, thereby capturing the lattice size and orientation of a given crystal. We conduct experiments on two commonly used material benchmark datasets, including the Materials Project [16] and Jarvis [8]. Results show that Matformer outperforms baseline methods consistently on various tasks. In addition, our results demonstrate the importance of both periodic invariance and periodic pattern encoding for crystal representation learning.

## 2    Background

**Crystal property prediction and crystal structures.** Given a crystal represented as $(\mathbf{A}, \mathbf{P}, \mathbf{L})$, crystal property prediction aims to predict a target property value $y$, which is either real as $y \in \mathbb{R}$ or categorical as $y \in \{1, 2, \cdots, C\}$, for regression or classification task with $C$ classes, respectively. Specifically, a crystal is represented as a unit cell with periodic patterns. A unit cell is a minimum repeatable structure for the given crystal, and it can be described by matrices $\mathbf{A}$ and $\mathbf{P}$. $\mathbf{A} = [\boldsymbol{a}_1, \boldsymbol{a}_2, \cdots, \boldsymbol{a}_n]^T \in \mathbb{R}^{n \times d_a}$ is the atom feature matrix, where $\boldsymbol{a}_i \in \mathbb{R}^{d_a}$ is the $d_a$-dimensional feature vector for atom $i$ in the unit cell. $\mathbf{P} = [\boldsymbol{p}_1, \boldsymbol{p}_2, \cdots, \boldsymbol{p}_n]^T \in \mathbb{R}^{n \times 3}$ is the position matrix, where $\boldsymbol{p}_i \in \mathbb{R}^3$ contains the Cartesian coordinates for atom $i$ in 3D space. To further encode periodic patterns, an additional lattice matrix $\mathbf{L} = [\boldsymbol{\ell}_1, \boldsymbol{\ell}_2, \boldsymbol{\ell}_3]^T \in \mathbb{R}^{3 \times 3}$ is used to describe how a unit cell repeats itself in three directions, including $\boldsymbol{\ell}_1, \boldsymbol{\ell}_2$, and $\boldsymbol{\ell}_3$. We show a 2D case of periodic patterns for easy illustration in Fig. 1 (a). Note that crystals usually possess irregular shapes in practice. Hence, $\boldsymbol{\ell}_1, \boldsymbol{\ell}_2$, and $\boldsymbol{\ell}_3$ are not always orthogonal in 3D space. Formally, given a crystal representation

$(\mathbf{A}, \mathbf{P}, \mathbf{L})$, the infinite crystal structure can be represented as

$$\hat{\mathbf{P}} = \{\hat{\boldsymbol{p}}_i | \hat{\boldsymbol{p}}_i = \boldsymbol{p}_i + k_1\boldsymbol{\ell}_1 + k_2\boldsymbol{\ell}_2 + k_3\boldsymbol{\ell}_3, \ k_1, k_2, k_3 \in \mathbb{Z}, i \in \mathbb{Z}, 1 \le i \le n\},$$
$$\hat{\mathbf{A}} = \{\hat{\boldsymbol{a}}_i | \hat{\boldsymbol{a}}_i = \boldsymbol{a}_i, i \in \mathbb{Z}, 1 \le i \le n\}. \tag{1}$$

Here, $\hat{\mathbf{P}}$ contains all possible positions for each atom $i$, associated with the same $\boldsymbol{a}_i$ in $\hat{\mathbf{A}}$.

**Multi-edge graph construction for crystals.** The multi-edge graph construction proposed by Xie and Grossman [52] aims to capture atom interactions across cell boundaries, which are imposed artificially. In a regular molecular graph, a node corresponds to a single atom. In contrast, in a multi-edge graph, node $i$ represents atom $i$ and all its duplicates in the infinite 3D space. Apparently, node $i$ contains the atom features vector $\boldsymbol{a}_i$ and all positions in the set $\{\hat{\boldsymbol{p}}_i | \hat{\boldsymbol{p}}_i = \boldsymbol{p}_i + k_1\boldsymbol{\ell}_1 + k_2\boldsymbol{\ell}_2 + k_3\boldsymbol{\ell}_3, \ k_1, k_2, k_3 \in \mathbb{Z}\}$. Formally, the multi-edge graph construction method builds edges between nodes as follows. Given a prefixed radius $r \in \mathbb{R}$, if there exists any 3-tuple $(k'_1, k'_2, k'_3)$, where $k'_1, k'_2, k'_3 \in \mathbb{R}$, such that the Euclidean distance $d'_{ji} \in \mathbb{R}$ satisfies $d'_{ji} = ||\boldsymbol{p}_j + k'_1\boldsymbol{\ell}_1 + k'_2\boldsymbol{\ell}_2 + k'_3\boldsymbol{\ell}_3 - \boldsymbol{p}_i||_2 \le r$, an edge is built from $j$ to $i$ with the initial edge feature $d'_{ji}$. An example of the edge construction is shown in Fig. 1 (b). Intuitively, if there exist $m$ positions of node $j$ within the radius of the center node $i$, this method builds $m$ edges from node $j$ to node $i$. By considering all possible positions of every node within a predefined radius in 3D space, the multi-edge graph construction method can in essence capture atom interactions across cell boundaries [40, 33, 4, 41, 6].

## 3 Periodic invariance and periodic pattern encoding for crystals

Different from molecular graphs, crystal graphs consist of a minimum unit cell repeating itself on a regular lattice in 3D space. When encoding such periodic structures, unique challenges lie in periodic invariance and periodic pattern encoding. In this section, we propose to formally define and analyze the importance of these two components.

### 3.1 Periodic invariance for crystals

Periodic invariance is proposed based on E(3) invariance, which is defined as below.

**Definition 1** (Unit Cell E(3) Invariance). *A function $f : (\mathbf{A}, \mathbf{P}, \mathbf{L}) \to \mathcal{X}$ is unit cell E(3) invariant such that for all $Q \in \mathbb{R}^{3 \times 3}, |Q| = \pm 1$ and $b \in \mathbb{R}^3$, we have $f(\mathbf{A}, \mathbf{P}, \mathbf{L}) = f(\mathbf{A}, Q\mathbf{P} + b, Q\mathbf{L})$, where Q is rotation and reflection transformations, and b is translation transformations in 3D space.*

Intuitively, the structure of a cell remains the same when either applying rotations and reflections to position matrix $\mathbf{P}$ and lattice matrix $\mathbf{L}$ together, or applying translations to $\mathbf{P}$ only. Correspondingly, the output of the unit cell E(3) invariant function should remain the same.

In addition to unit cell E(3) invariance, periodic invariance is also shown necessary for generating valid crystal representations. Specifically, when the lattice matrix $\mathbf{L} = [\boldsymbol{\ell}_1, \boldsymbol{\ell}_2, \boldsymbol{\ell}_3]^T \in \mathbb{R}^{3 \times 3}$ is fixed for a crystal, we can still obtain different position matrices $\mathbf{P} \in \mathbb{R}^{n \times 3}$ and different unit cell structures by shifting the period boundaries. As shown in Fig. 2 (a) and (b), the formed unit cell structures are different for the same crystal by shifting period boundaries. To this end, we further introduce periodic invariance, which shows that when the periodic boundaries are shifted

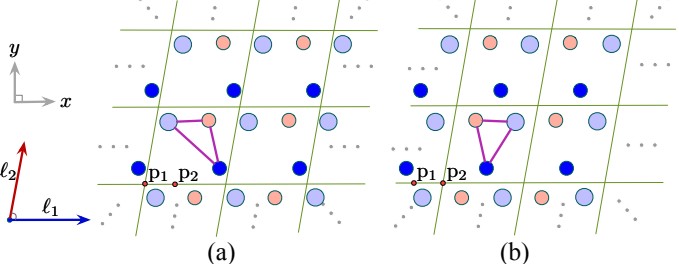

Figure 2: Illustration of periodic invariance. Purple lines are the edges between nodes inside a unit cell. Red points are the corner points of the unit cells. For example, $p_1$ and $p_2$ are for unit cells in (a) and (b), respectively. (a) and (b) show different unit cells describing the same crystal, caused by shifting the period boundaries along $x$ axis from $p_1$ to $p_2$. By comparing (a) and (b), we show a graph construction method that breaks periodic invariance.

or scaled up, the periodic invariant representation should remain the same. Formally, based on Sec. 2, we further define a function $\Phi : (\hat{\mathbf{A}}, \hat{\mathbf{P}}, \mathbf{L}, p) \to (\mathbf{A}, \mathbf{P})$ simulating how to form different unit cells

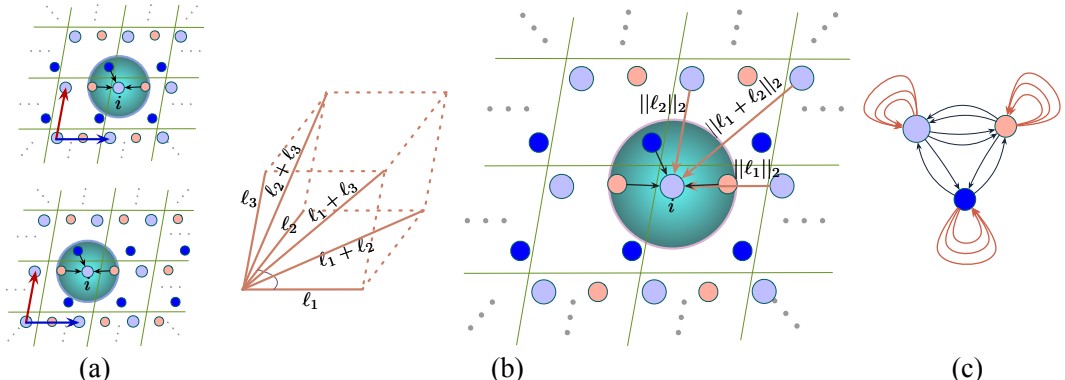

(a)           (b)           (c)

Figure 4: Illustrations of the used radius-based graph construction method and the proposed periodic pattern encoding for Matformer. Black arrows are the formed edges by radius-based graph construction and brown arrows are self-connecting edges. (a). Illustration that the used radius-based graph construction satisfies periodic invariance. (b). Illustration of periodic pattern encoding using self-connecting edges. On the left, we show the designed self-connecting edges to encode the lattice matrix $\mathbf{L} = [\boldsymbol{\ell}_1, \boldsymbol{\ell}_2, \boldsymbol{\ell}_3]^T \in \mathbb{R}^{3 \times 3}$ in 3D space. We design six self-connecting edges $||\boldsymbol{\ell}_1||_2$, $||\boldsymbol{\ell}_2||_2$, $||\boldsymbol{\ell}_3||_2$, $||\boldsymbol{\ell}_1 + \boldsymbol{\ell}_2||_2$, $||\boldsymbol{\ell}_1 + \boldsymbol{\ell}_3||_2$, $||\boldsymbol{\ell}_2 + \boldsymbol{\ell}_3||_2$. The geometric shape of $\mathbf{L}$ can be determined by these six edges. On the right, we show the added self-connecting edges in 2D space for easy illustration. (c). Illustration of the constructed graph with periodic pattern encoding for the above 2D case.

from a given infinite crystal structure. For an infinite crystal structure represented as $(\hat{\mathbf{A}}, \hat{\mathbf{P}})$, $\Phi$ uses a corner point $\mathrm{p}$ and shape matrix $\mathbf{L}$ to form a unit cell represented as $(\mathbf{A}, \mathbf{P})$. In addition, we use $\boldsymbol{\alpha} \in \mathbb{N}_+^3$ to indicate the scaling up of a repeating unit cell formed by periodic boundaries. Then the formal definition of periodic invariance is below.

**Definition 2** (Periodic Invariance). *A unit cell E(3) invariant function $f : (\mathbf{A}, \mathbf{P}, \mathbf{L}) \to \mathcal{X}$ is periodic invariant if $f(\mathbf{A}, \mathbf{P}, \mathbf{L}) = f(\Phi(\hat{\mathbf{A}}, \hat{\mathbf{P}}, \boldsymbol{\alpha}\mathbf{L}, \mathrm{p}), \boldsymbol{\alpha}\mathbf{L})$ holds for all $\mathrm{p} \in \mathbb{R}^3$ and $\boldsymbol{\alpha} \in \mathbb{N}_+^3$.*

**Significance of periodic invariance.** Breaking periodic invariance will result in different crystal graphs for the same crystal. In Fig. 2 (a) and (b), we show a graph construction method that breaks periodic invariance. This method is employed by a transformer-based model, known as Graphormer [54], which first uses radius to include all the atoms of interest in nearby cells and then builds a fully connected graph. A detailed illustration is shown in Appendix. A.1. Based on the formal definition of periodic invariance, in this study, we aim to integrate such an important component in our Matformer. By doing this, our model is able to construct a distinct crystal graph for a given crystal structure, resulting in a more informative and discriminative crystal learning scheme.

## 3.2 Periodic pattern encoding

As introduced in Sec. 2, $\mathbf{L} = [\boldsymbol{\ell}_1, \boldsymbol{\ell}_2, \boldsymbol{\ell}_3]^T \in \mathbb{R}^{3 \times 3}$ containing periodic patterns is another key component to describe crystal structures. Essentially, periodic patterns show how the minimum repeatable structure $(\mathbf{A}, \mathbf{P})$ expands itself in infinite 3D space. Without such periodic pattern encoding in $\mathbf{L}$, crystal structures are treated as finite structures similar to molecules. As shown in Fig. 3, the widely used multi-edge graph construction method [52, 4, 41, 40, 6] only captures local interactions among atoms but ignores the important periodic patterns. However, such periodic repeating patterns need to be captured explicitly, as lattices of different sizes and orientations may correspond to different materials. Hence, To better represent the infinite structures of crystals, we argue that the periodic patterns

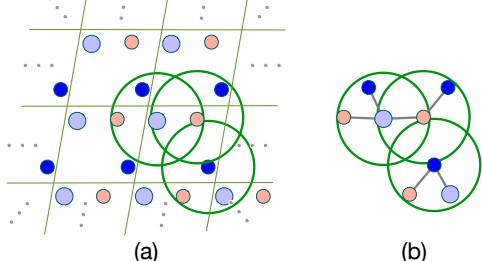

(a)        (b)

Figure 3: Illustration that periodic patterns are not encoded in a multi-edge graph. Grey lines show the captured topology information by the multi-edge graph construction. We use three radius circles because there are three atoms in a unit cell. (a). Illustration of the multi-edge graph construction. (b). Illustration that the multi-edge graph method only captures local geometric information but ignores periodic patterns for the infinite structure.

$\mathbf{L} \in \mathbb{R}^{3 \times 3}$ should be explicitly taken into consideration in crystal learning.

# 4 The proposed Matformer

## 4.1 The proposed graph construction methods

In this section, we introduce the proposed graph construction methods for Matformer. Our methods effectively integrate periodic invariance and periodic pattern encoding.

**Invariant crystal graph construction**. We consider two crystal graph construction methods, including radius-based graph construction and fully connected graph construction. Both of them satisfy periodic invariance, and mathematical proofs can be found in Appendix. A.2. In Fig. 2, we show by example that treating atoms as single nodes breaks periodic invariance. Instead, our methods follow the fashion introduced in Sec. 2 to treat node $i$ as atom $i$ and all its repeated duplicates.

We use the multi-edge graph construction [52] introduced in Sec. 2 as an alternative crystal graph construction method. Note that although the multi-edge graph construction satisfies periodic invariance as in Fig. 4 (a), several existing works [52, 33] do not follow the settings exactly thus breaking periodic invariance. Within the given radius shown in Fig. 4 (a), these studies form the neighborhood of node $i$ by selecting $t$ nearest neighbors ranked by geometric distances. If there exist several different atoms with the same distance to node $i$, there is no deterministic way to select from them, as shown in Appendix. A.1. As a result, periodic invariance cannot be guaranteed.

Given the fully-connected fashion employed in Graphormer has achieved impressive performance on molecular learning, we further propose another graph construction method for Matformer, known as the fully connected graph construction. This method uses a different strategy to determine neighbors for each center node. Specifically, for node $i$ and $j$, it builds edges for the entries corresponding to the $t$ smallest distances in $\{d_{ij}|d_{ij} = ||\boldsymbol{p}_i - \boldsymbol{p}_j + k_1^{'}\boldsymbol{\ell}_1 + k_2^{'}\boldsymbol{\ell}_2 + k_3^{'}\boldsymbol{\ell}_3||_2, \ k_1^{'}, k_2^{'}, k_3^{'} \in \mathbb{Z}\}$. It can be seen every pair $i$ and $j$ is connected in the constructed graph and there are $t$ edges between them.

Overall, the used multi-edge graph construction and proposed fully-connected graph construction both satisfy the important periodic invariance. Particularly, they possess great flexibility to be used in future studies for crystal learning. In this study, we employ both methods as part of our proposed Matformer, and in main experiments, the radius-based method is used due to better empirical performance.

**Periodic pattern encoding with self-connecting edges**. In this study, we propose to encode the important $\mathbf{L} = [\boldsymbol{\ell}_1, \boldsymbol{\ell}_2, \boldsymbol{\ell}_3]^T \in \mathbb{R}^{3\times3}$ into crystal graphs by adding self-connecting edges. As mentioned in Sec. 3.2, periodic patterns describe sizes and orientations of lattices of a crystal structure, eventually determining the properties of this crystal. A natural step for encoding such repeating periodic patterns is to consider the relative positions between an atom and its nearby repeated duplicates. Formally, given an atom $i$ with position $\boldsymbol{p}_i$ and $\mathbf{L} = [\boldsymbol{\ell}_1, \boldsymbol{\ell}_2, \boldsymbol{\ell}_3]^T \in \mathbb{R}^{3\times3}$, we need to encode the atom's three nearby duplicates with positions $\boldsymbol{p}_i + \boldsymbol{\ell_1}$, $\boldsymbol{p}_i + \boldsymbol{\ell_2}$, and $\boldsymbol{p}_i + \boldsymbol{\ell_3}$. It is widely known that a direction vector $\boldsymbol{\ell}_i$ is determined by both its length $||\boldsymbol{\ell}_i||_2$ and orientation. Essentially, $||\boldsymbol{\ell}_i||_2$ indicates the geometric distance between atom $i$ and its corresponding duplicate. However, the computing of orientation information, such as angles, usually induces high complexity. Hence, it is not practical to encode such orientation information into transformer architectures. To this end, we propose to use geometric distances solely to implicitly consider the orientation information.

Specifically, we use additional distances to determine angles between any two direction vectors in $\mathbf{L}$. For example, the angle between $\boldsymbol{\ell}_1$ and $\boldsymbol{\ell}_2$ can be easily computed by $||\boldsymbol{\ell}_1||_2$, $||\boldsymbol{\ell}_2||_2$, and an additional distance $||\boldsymbol{\ell}_1 + \boldsymbol{\ell}_2||_2$. Hence, based on these three distances, we can determine lengths of $\boldsymbol{\ell}_1$ and $\boldsymbol{\ell}_2$, and the relative orientation between them. Extensively, we use six geometric distances, including $||\boldsymbol{\ell}_1||_2, ||\boldsymbol{\ell}_2||_2, ||\boldsymbol{\ell}_3||_2, ||\boldsymbol{\ell}_1 + \boldsymbol{\ell}_2||_2, ||\boldsymbol{\ell}_2 + \boldsymbol{\ell}_3||_2$, and $||\boldsymbol{\ell}_1 + \boldsymbol{\ell}_3||_2$ in our study, as shown in Fig. 4 (b). By doing this, the length of each direction vector and the angle between any two direction vectors can all be determined. As a result, the shape formed by lattice matrix $\mathbf{L}$ is then fixed. Overall, we build the aforementioned six geometric distances as six self-connecting edges for node $i$. By doing this, our model is capable of encoding periodic patterns in $\mathbf{L}$ completely, resulting in a more accurate crystal representation learning scheme. Importantly, our approach also guarantees periodic invariance.

Overall, the graph construction for Matformer consists of two necessary stages, including invariant graph construction and periodic pattern encoding. For the first stage, we rigorously prove that the multi-edge graph construction satisfies periodic invariance in Appendix. A.2 and show that several previous works [52, 33] break periodic invariance using a different neighbor selection strategy. Additionally, we propose a fully-connected crystal graph construction method satisfying periodic

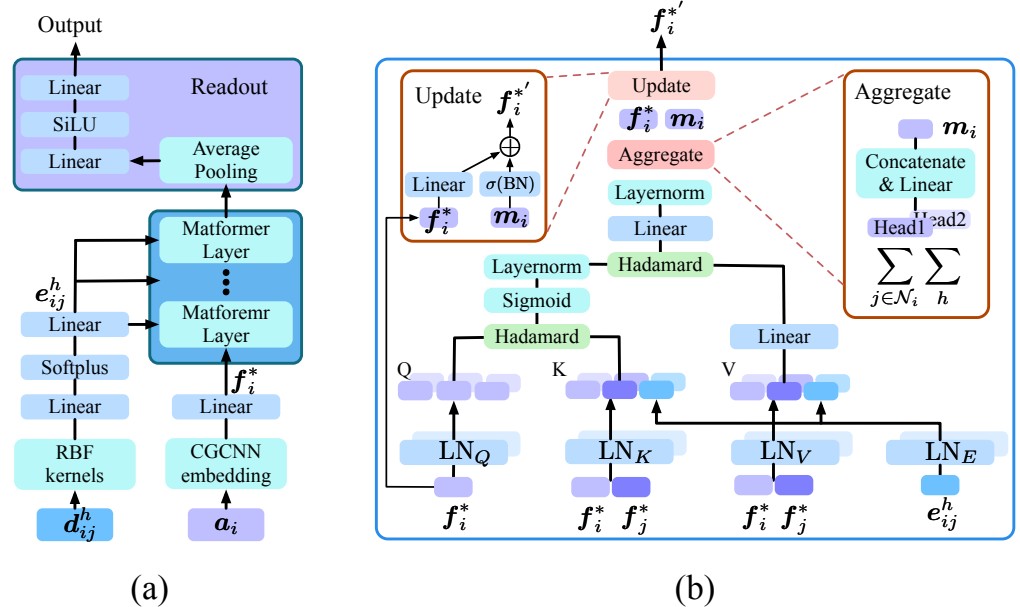

Figure 5: Illustration of detailed Architecture of Matformer. The overlapping graphics are used to denote two different attention heads. (a). Illustration of Matformer pipeline. (b). Illustration of the detailed Matformer layer in (a). We show the case with two attention heads for simplicity.

invariance, and the method could be used in future studies for crystal representation learning. For the second stage, we naturally encode periodic patterns into constructed graphs by adding self-connecting edges without breaking periodic invariance. A constructed graph in 2D case is shown in Fig. 4 (c).

## 4.2 Message passing scheme

Building on constructed graphs introduced in Sec. 4.1, we propose our message passing scheme for Matformer. Formally, we denote a constructed crystal graph as $G = (\mathbf{A}, \mathbf{E})$. Here, each $\boldsymbol{a}_i \in \mathbf{A}$ is the $d_a$-dimensional feature vector for atom $i$, as introduced in Sec. 2. Particularly, $\boldsymbol{e}_{ij}^h \in \mathbf{E}$ is $d_e$-dimensional feature vector for the $h$-th edge between nodes $i$ and $j$. We follow the regular attention mechanism that computes query, key, and value [30, 28]. Our proposed message passing scheme is composed of three steps; those are, edge-wise attention coefficients computing, edge-wise value message computing, and node updating. Formally, we let $\boldsymbol{f}_i^{\star\ell}$ denote the input feature vector of node $i$ for the $\ell$-th layer of Matformer. The message passing scheme of the $\ell$-th layer is described as below.

In the first step, $\boldsymbol{q}_{ij}^h$, $\boldsymbol{k}_{ij}^h$ and $\boldsymbol{\alpha}_{ij}^h$ for the $h$-th edge between $i$ and $j$ are computed as

$$\boldsymbol{q}_i = \text{LN}_Q(\boldsymbol{f}_i^{\star\ell}), \ \boldsymbol{k}_i = \text{LN}_K(\boldsymbol{f}_i^{\star\ell}), \ \boldsymbol{k}_j = \text{LN}_K(\boldsymbol{f}_j^{\star\ell}), \ \boldsymbol{e}_{ij}^{h'} = \text{LN}_E(\boldsymbol{e}_{ij}^h),$$

$$\boldsymbol{q}_{ij}^h = (\boldsymbol{q}_i|\boldsymbol{q}_i|\boldsymbol{q}_i), \ \boldsymbol{k}_{ij}^h = (\boldsymbol{k}_i|\boldsymbol{k}_j|\boldsymbol{e}_{ij}^{h'}), \ \boldsymbol{\alpha}_{ij}^h = \frac{\boldsymbol{q}_{ij}^h \circ \boldsymbol{k}_{ij}^h}{\sqrt{d_{\boldsymbol{k}_{ij}^h}}}, \tag{2}$$

where $\text{LN}_Q$, $\text{LN}_K$, and $\text{LN}_E$ denote the linear transformations to compute query, key, and edge embedding in $\ell$-th layer, respectively. $\boldsymbol{e}_{ij}^{h'}$ is the intermediate output for $\boldsymbol{e}_{ij}^h$. We use $\circ$ and $|$ to denote Hadamard Product and concatenation. Note that $\boldsymbol{q}_{ij}^h$ is the concatenation of three $\boldsymbol{q}_i$ vectors to match the dimension of $\boldsymbol{k}_{ij}^h$. By doing this, when computing $\boldsymbol{\alpha}_{ij}^h$, $\boldsymbol{q}_i$ attends each of $\boldsymbol{k}_i$, $\boldsymbol{k}_j$, and $\boldsymbol{e}_{ij}^{h'}$ for integrating more information in attention.

Particularly, we omit the softmax to enhance the model's capability to distinguish nodes with different degrees, and to make the whole network more efficient.

After obtaining coefficients $\boldsymbol{\alpha}_{ij}^h$, in the second step, we compute $\boldsymbol{m}_{ij}^h$ that is the message of $\boldsymbol{e}_{ij}^h$ as

$$\boldsymbol{v}_i = \text{LN}_V(\boldsymbol{f}_i^{\star\ell}), \ \boldsymbol{v}_j = \text{LN}_V(\boldsymbol{f}_j^{\star\ell}), \boldsymbol{m}_{ij}^h = \text{sigmoid}(\text{LNorm}(\boldsymbol{\alpha}_{ij}^h)) \circ \text{LN}_{\text{update}}(\boldsymbol{v}_i|\boldsymbol{v}_j|\boldsymbol{e}_{ij}^{h'}), \tag{3}$$

where $LN_V$ and $LN_{update}$ are the linear transformations to compute value and the updated message, and LNorm denotes the layer normalization operation.

Finally, in the third step, we compute node $i$'s feature vector $\boldsymbol{f}_i^\ell$. Specifically, we first obtain message $\boldsymbol{m}_i$ by aggregating information from node $i$'s neighborhood over multiple edges, then achieve $\boldsymbol{f}_i^\ell$ as

$$\boldsymbol{m}_i = \sum_{j \in \mathcal{N}_i} \sum_h \text{LNorm}(\text{LN}_{\text{msg}}(\boldsymbol{m}_{ij}^h)), \ \boldsymbol{f}_i^\ell = \text{LN}_{\text{fea}}(\boldsymbol{f}_i^{\star\ell}) + \sigma(\text{BN}(\boldsymbol{m}_i)), \qquad (4)$$

where $\sigma$ is the used activation function, and BN indicates batch normalization. In addition, $\text{LN}_{\text{msg}}$ and $\text{LN}_{\text{fea}}$ are linear transformations to update the messages on edges and the old atom features.

Graphormer represents an effective transformer variant for molecular graph learning. The differences between Graphormer and the proposed Matformer lie in both graph construction and message passing scheme. Firstly, Graphormer treats every atom as a single node and breaks periodic invariance, as mentioned in Sec. 3.1. For the message passing, Graphormer uses the node-wise attention and encodes pairwise distances as attention bias. It cannot work properly on multi-edge graphs for crystals. While Matformer is specifically designed for multi-edge crystal graphs by performing edge-wise attention and encoding geometric information into edge-wise messages, as described above. The detailed architecture of our Matformer is shown in Fig. 5.

## 5   Related work

**Crystal property prediction**. Several existing methods [46, 18, 19, 14] model crystals as chemical formulas and employ sequence models to process them. Other studies [52, 40, 33, 4, 41, 6] consider 3D structures and formulate crystals as 3D graphs, then apply GNNs to learn from crystal graphs. As crystals are essentially periodically repeated structures, the graph construction needs to consider periodic invariance and periodic pattern encoding. There are limited efforts to identify these two unique components. As an early work, CGCNN [52] proposes to capture atom interactions across artificial cell boundaries by using multi-edge graphs described in Sec. 2. The multi-edge graph satisfies periodic invariance as described in Sec. 3.1, but fails to consider the important periodic patterns, as described in Sec. 3.2. The multi-edge graph construction method is widely used in the following studies [40, 33, 4, 41, 6, 1]. Based on the constructed crystal graphs, many GNN variants have been proposed for effective crystal representation learning [52, 40, 33, 4, 41, 6, 1]. Specifically, Nequip [1] considers E(3) equivariance for materials, and satisfies periodic invariance using multi-edge graphs, but fails to capture periodic repeating patterns. We also notice a recent work [48] for periodic graph generation, which considers periodic graphs as finite graphs and breaks periodic invariance. Recently, ALIGNN [6] achieves the best performance on two major material datasets. It uses angle information in the message passing to generate more informative and discriminative representations. However, the use of angles introduces excessive time complexity.

**Geometric GNNs and graph transformer**. Many efforts have been made to incorporate 3D geometric information in molecular learning. Exemplary studies include SchNet [41], DimeNet [23, 22], SphereNet [31], GemNet [24], etc. However, these methods are designed for molecules without periodic patterns. Recently, graph transformers [54] using geometric information, e.g., Graphormer [54], have shown great potential on real-world graph data. However, Graphormer considers neither periodic invariance nor periodic pattern encoding.

**Differences with our method.** To the best of our knowledge, periodic invariance and periodic pattern encoding described in Sec. 3 are rarely identified and explored in existing works for crystal property prediction. CGCNN [52] breaks periodic invariance on some corner cases because it uses twelve nearest neighbors determined only by distances as described in Sec. 4.1. In addition, previous methods including CGCNN [52], SchNet [41], MEGNET [4], CYATT [40], GATGNN [33], NEQUIP [1] and ALIGNN [6], all fail to consider the important periodic pattern encoding as introduced in Sec. 3.2. Especially, following GAT [45], GATGNN [33] employs a very limited kind of attention mechanism that is not conditioned on query, as explained in GATv2 [2]. As a result, the model capacity is reduced compared with the self attention mechanism employed in Matformer. In addition, the usage of softmax limits the capability of GATGNN of distinguishing nodes with different degrees, as mentioned in Sec. 4.2. For Graphormer [54], although it achieved remarkable success on the Open Catalyst Challenge [3], the employed graph construction method breaks periodic invariance when

Table 1: Comparison in terms of test MAE on The Materials Project dataset. To make the comparison clear and fair, We show results from retrained models using exactly the same training, validation, and test sets. Results from original papers are shown in Appendix A.5. The best results are shown in **bold** and the second best results are shown with underlines.

| Method | Formation Energy eV/atom | Band Gap eV | Bulk Moduli log(GPa) | Shear Moduli log(GPa) |
|---|---|---|---|---|
| CGCNN [52] | 0.031 | 0.292 | 0.047 | 0.077 |
| SchNet [41] | 0.033 | 0.345 | 0.066 | 0.099 |
| MEGNET [4] | 0.030 | 0.307 | 0.060 | 0.099 |
| GATGNN [33] | 0.033 | 0.280 | 0.045 | 0.075 |
| ALIGNN [6] | 0.022 | 0.218 | 0.051 | 0.078 |
| Matformer | **0.021** | **0.211** | **0.043** | **0.073** |

applied to crystals. Compared with Graphormer, Matformer is specifically designed for crystals considering both periodic invariance and periodic patterns.

# 6 Experimental studies

## 6.1 Experimental setup

We conduct experiments on two material benchmark datasets, including The Materials Project [16] and JARVIS [8]. The detailed descriptions for The Materials Project and JARVIS datasets are shown in Appendix. A.3. Baseline methods include CFID [7], CGCNN [52], SchNet [41], MEGNET [4], GATGNN [33], and ALIGNN [6]. Unless otherwise specified, for all the baseline methods, we report the results taken from the referred papers or provided by original authors. All Matformer models are trained using the Adam optimizer [21] with weight decay [32] and one cycle learning rate scheduler [43]. We only slightly adjust learning rates from 0.001 and training epochs from 500 for different tasks. Detailed Matformer configurations for different tasks are provided in Appendix. A.4.

## 6.2 Experimental results

**The Materials Project.** We first use The Materials Project-2018.6.1 dataset [4], which contains 69239 crystals, to evaluate Matformer. We notice that previous works [52, 41, 4, 33, 6] compare with each other either using datasets of different sizes, or using datasets with the same size but splitting the datasets with different random seeds. To make the comparison clear and fair, we retrain all corresponding models using exactly the same training, validation and test sets across all methods and report the results in Table. 1. To avoid confusion, we still put original results from referred papers in Table. 1 inside parentheses. For retrained baseline models, we provide the detailed configurations in Appendix. A.5. The used metric is test MAE following previous studies [52, 41, 4, 33, 6].

It can be seen from Table. 1 that Matformer achieves the best performances on all tasks consistently by significant margins. Specifically, it reduces the formation energy by 4.5% of the second best model, which is a significant margin. Furthermore, for Bulk Moduli and Shear Moduli tasks with only 4664 training samples, Matformer achieves the best performances, indicating Matformer's adaptive ability to tasks of small training scales.

**JARVIS dataset**. The quantitative results for Jarvis are shown in Table. 2. Matformer outperforms the baseline methods significantly on all of these five tasks. Compared with ALIGNN, Matformer has stronger discriminative ability due to explicit encoding of periodic patterns. Specifically, Matformer reduces Jarvis Ehull by 0.012, which is 15.8% of ALIGNN. Furthermore, Matformer achieves the best performances for Bulk Moduli and Shear Moduli in the Mateirals Project with 4664 training samples, and Bandgap(MBJ) in JARVIS with 14537 training samples, indicating its adaptive ability to tasks of various data scales. Overall, the superior performances show the effectiveness of periodic pattern encoding in our Matformer message passing. In addition, compared with ALIGNN, our Matformer is more efficient. We evaluate the efficiency of Matformer by comparing with ALIGNN using JARVIS formation energy dataset. The mean time of ten runs for training and inference using best model configurations of ALIGNN and Matformer are reported. We also report the total number of parameters of each model. In Table. 3, we show that Matformer is three times faster than ALIGNN

Table 2: Comparison between Matformer and other baselines in terms of test MAE on JARVIS dataset. The best results are shown in **bold** and the second best results are shown with underlines.

| Method | Formation Energy eV/atom | Bandgap(OPT) eV | Total Energy eV/atom | Ehull eV | Bandgap(MBJ) eV |
|---|---|---|---|---|---|
| CFID [7] | 0.14 | 0.30 | 0.24 | 0.22 | 0.53 |
| CGCNN [52] | 0.063 | 0.20 | 0.078 | 0.17 | 0.41 |
| SchNet [41] | 0.045 | 0.19 | 0.047 | 0.14 | 0.43 |
| MEGNET [4] | 0.047 | 0.145 | 0.058 | 0.084 | 0.34 |
| GATGNN [33] | 0.047 | 0.17 | 0.056 | 0.12 | 0.51 |
| ALIGNN [6] | 0.0331 | 0.142 | 0.037 | 0.076 | 0.31 |
| Matformer | **0.0325** | **0.137** | **0.035** | **0.064** | **0.30** |

Table 3: Efficiency comparison with ALIGNN on Jarvis Formation Energy task. We show the training time per epoch, total training time, inference time for the whole test set, and total number of parameters.

| Models | Time/epoch | Total | Inference | Model Para. |
|---|---|---|---|---|
| ALIGNN | 327 s | 27.3 h | 156 s | 15.4 MB |
| Matformer | 64 s | 8.9 h | 59 s | 11.0 MB |

in total training time and near three times faster in inference time, for the whole test set. Matformer is also much lighter than ALIGNN in terms of model size.

**Energy within Threshold**. Following OC20 [3], we use energy within threshold (EwT), which measures the percentage of estimated energies that are likely to be practically useful when the absolute error is within a certain threshold, to evaluate Matformer's capability for periodic graph learning. This metric is new, but is well recognized by the community as it is useful in practice. Due to significant performance gaps between ALIGNN and other baseline methods on formation energy and total energy for these two datasets in terms of mean absolute error (MAE), we only compare Matformer with ALIGNN. Table. 4 shows that Matformer outperforms ALIGNN consistently for all three energy prediction tasks. Interestingly, the performance gains of our Matformer beyond ALIGNN in terms of EwT mainly come from more accurate energy predictions within absolute error of 0.01. Compared with JARVIS, the Materials Project has 15422 more traning samples. As a result, the percentage of predicted energies obtained by Matformer within 0.01 increases by 14.69%, which is much better than ALIGNN, revealing the huge potential of Matformer when larger crystal dataset is available.

## 6.3 Ablation studies

In this section, we demonstrate the importance of periodic invariance and explicit repeating pattern encoding for crystal representation learning by ablation studies. We also evaluate the building blocks particularly designed for our Matformer. Specifically, we conduct experiments on JARVIS formation energy, and the test MAE is used as the quantitative evaluation metric. We also provide ablation studies on the use of sigmoid and layernorm instead of softmax in Matformer layer in Appendix A.6.

**Periodic invariant graph construction**. We demonstrate the importance of periodic invariant graph construction by comparing radius multi-edge graph, denoted as Radius, with the graph construction method proposed by Graphormer, denoted as OCgraph, on the exactly same Matformer architecture. Note that the constructed crystal graphs by OCgraph are super large, containing more than $n^2$ edges,

Table 4: Comparison between Matformer and ALIGNN in terms of EwT on JARVIS Formation Energy, JARVIS Total Energy and The Materials Project Formation Energy. We use EwT (0.02) to mark the threshold of 0.02 and EwT (0.01) to mark the threshold of 0.01. The best results are in **bold**.

| Method | Formation MP EwT (0.01) | Formation MP EwT (0.02) | Formation JARVIS EwT (0.01) | Formation JARVIS EwT (0.02) | Total JARVIS EwT (0.01) | Total JARVIS EwT (0.02) |
|---|---|---|---|---|---|---|
| ALIGNN | 49.94% | 71.10% | 39.59% | 59.64% | 35.09% | 55.20% |
| Matformer | **55.86%** | **75.02%** | **41.17%** | **60.25%** | **36.84%** | **57.36%** |

Table 5: Ablation studies on periodic invariance and periodic pattern encoding. We use OCgraph to denote graph construction method proposed by Graphormer. PI denotes periodic invariance and PE denotes periodic encoding.

| Graph | PI | PE | layer | head | batch | Test MAE |
|-------|----|----|-------|------|-------|----------|
| OCgraph | × | × | 3 | 1 | 32 | 0.0530 |
| Radius w/o PE | ✓ | × | 3 | 1 | 32 | 0.0348 |
| Radius w/o PE | ✓ | × | 5 | 4 | 64 | 0.0337 |
| T-fully w PE | ✓ | ✓ | 5 | 4 | 64 | 0.0402 |
| Radius w PE | ✓ | ✓ | 5 | 4 | 64 | **0.0325** |

where $n$ is the atom number in a cell. We adjust the Matformer configurations to train these large graphs on a single RTX A6000 GPU. It can be seen from Table. 5 that when using OCgraph to our Matformer, the test MAE drops dramatically of 53% because of breaking periodic invariance, compared with radius-based multi-edge graphs in Matformer. We also compare two periodic invariant graph construction methods described in Sec. 4.1. We denote fully connected graph with $t$ smallest pairwise distances as T-fully, and use $t = 3$. The result shows that Radius is better than T-fully.

**Encoding of repeating patterns**. We denote periodic pattern encoding as PE. In Table. 5, we show that omitting the periodic pattern encoding results in a significant drop of test MAE from 0.0325 to 0.0337, revealing the importance of periodic patterns for crystal representation learning.

**Complexity of introducing angular information.** Dropping angular information largely improves running efficiency of Matformer compared with ALIGNN. We show that for the original crystal graph with $n$ nodes and $6n$ edges, the corresponding line graph with angles will have $6n$ nodes and $66n$ edges, leading to high computational cost. The detailed complexity

Table 6: Ablation studies on angular information. We show the training time per epoch, total training time, and test MAE.

| Models | MAE | Time/epoch | Total |
|--------|-----|-----------|-------|
| Matformer | .0325 | 64 s | 8.9 h |
| Matformer + Angle SBF | .0332 | 173 s | 23.8 h |
| Matformer + Angle RBF | .0325 | 165 s | 22.9 h |

analysis of adding angular information are provided in Appendix. A.7. Additionally, we provide the running time and performance analysis of Matformer with angle information in Table. 6. We use two Matformer layers to process extra angular information, and use Radial Basis Function kernels [6] and Spherical Bessel Functions with Spherical Harmonics [23, 31, 24] to encode angles, denoted as Matformer + Angle RBF and Matformer + Angle SBF. Table. 6 shows that introducing angular information will increase both the training time per epoch and in total by around 3 times, without much performance gain. This may due to the periodic invariant graph construction and periodic patterns encoding in Matformer already capture sufficient information to identify crystal structures.

# 7 Conclusions and discussions

In this work, we first propose to formally define periodic invariance and periodic pattern encoding for periodic graph learning. We then propose Matformer for periodic graph representation learning, which is invariant to periodicity and can capture repeating patterns explicitly. Experimental results on common benchmark datasets show that our Matformer outperforms baseline methods consistently. In addition, our results demonstrate the importance of periodic invariance and explicit periodic pattern encoding for crystal representation learning. One potential direction beyond this work is to include angular information properly to satisfy both periodic invariance and to encode periodic patterns with relatively low time complexity, and this is one limitation of our work. Besides, negative societal impacts of material discovery may apply to our work.

# Acknowledgments and Disclosure of Funding

We thank Tian Xie for answering our questions on CGCNN. This work was supported in part by National Science Foundation grant IIS-2006861.

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
