# OpenReview forum: "Periodic Graph Transformers for Crystal Material Property Prediction"
_NeurIPS.cc/2022/Conference — NeurIPS 2022 Accept_

### Official Review · Reviewer_oomT · 2022-07-09

**Rating:** 7
**Confidence:** 3
**Soundness:** 3 good
**Presentation:** 3 good
**Contribution:** 3 good

**Summary:**

Periodic graph is ubiquitous in the real world, such as crystal material. Representation learning on periodic graphs is a significant task for downstream tasks such as property prediction. This paper proposes a new transformer-based framework for the representation learning on periodic graphs. Specifically, it comes up with a new strategy to construct graphs and a way to encode lattice matrix L. The experiments section shows the effectiveness of the method.

**Questions:**

I would suggest the author address the questions mentioned under the "Weakness" in the "Strengths And Weaknesses" section of the review.

**Limitations:**

The limitations of the model are not mentioned, the negative societal impacts of your work are marked as N/A.

**Strengths And Weaknesses:**

Strength
1. The paper formularizes the periodic graph in a clear way using A, P and L matrix.
3. The experimental results show the effectiveness of the proposed method.

Weakness:
1. The motivation of the paper can be further improved:
    a) Why do we need to construct a graph from the unit-cell coordinates rather than directly using coordinates to conduct representation learning? What's the benefit to borrow the graph structure here? I believe there are a few works for representation learning on 3D positions such as [1], [2] and [3]
    b) What is the motivation to form self-connecting edges to contain the information of lattice matrix? Why not just concatenate it with the representation of a single unit cell?
2. It seems that section 3.1 and 3.2 can be moved to section 2 or they can just serve as a single section, since it seems that they are the motivation/theoretical ground of the proposed model not the model itself.
3. Line 45, the paragraph is titled as "Crystal property prediction" but it seems that the whole paragraph is about the definition of peridic lattice. The title might be modified accordingly.
4. It seems that E(3) invariance has been mentioned even in the abstract but it is formally defined in section 3.1.
5. Line 87, it mentions that "the structure f a cell remains the same when ...". Does the structure here mean the the output of the "f" function mentioned in the Definition 1 above?
6. Line 90, It says "periodic invariance is also shown necessary for generating valid crystal representation". Why?
7. In table 2 the paper compares the time complexity of the proposed method with ALIGNN. It would be great if some complexity analysis is provided here.
8. Some relevant works [4, 5] regarding the periodic graphs generation can be discussed in the related works section.
9. The code is not published for now to reproduce the results.


[1] Xu, Minkai, et al. "Geodiff: A geometric diffusion model for molecular conformation generation." arXiv preprint arXiv:2203.02923 (2022).
[2] Kim, Seohyun, Jaeyoo Park, and Bohyung Han. "Rotation-invariant local-to-global representation learning for 3d point cloud." Advances in Neural Information Processing Systems 33 (2020): 8174-8185.
[3] Court, Callum J., et al. "3-D inorganic crystal structure generation and property prediction via representation learning." Journal of chemical information and modeling 60.10 (2020): 4518-4535.
[4] Xie, Tian, et al. "Crystal diffusion variational autoencoder for periodic material generation." arXiv preprint arXiv:2110.06197 (2021).
[5] Wang, Shiyu, Xiaojie Guo, and Liang Zhao. "Deep Generative Model for Periodic Graphs." arXiv preprint arXiv:2201.11932 (2022).

---

> ### Author Response · Authors · 2022-08-02
> **Provided complexity analysis when introducing angular information; added related works; added code to supplementary material; added negative societal impacts and limitations. (3rd Part)**
>
> > 7. In table 2 the paper compares the time complexity of the proposed method with ALIGNN. It would be great if some complexity analysis is provided here.
>
> Thank you for bringing this question up.
>
> - Firstly, following your suggestion, we **provided the complexity analysis of introducing angular information** in crystal graphs in section 6.3 in the paper and Appendix A.6.  Assume that we have *n* atoms in a single cell and thus *n* nodes in the original graph. Also assume that every node has at least 12 neighbors (following the graph construction method of Matformer) and there are no self-connecting edges. This will result in a graph G=(V,E) where |V| = n and |E| = 12/2 n = 6n. When converting graph G into L(G), every edge is treated as a node in the line graph. So we have 6n nodes in the line graph. Every edge in the original graph is connecting 2 nodes and every node has (12 - 1) other edges, resulting in 22 neighboring edges for each edge. So we have 22*6n/2=66n edges in the converted line graph. **Compared with the original graph with |V| = n and |E| = 6n, the converted line graph with angle information is super large with the node number of 6n and edge number of 66n, which would induce high computational cost.**
>
> - Moreover, different from molecular graphs, atom number *n* can be super large for crystal graphs. To be concrete, the **maximum number of atoms in a single cell for the Materials Project is 296**. Also, we **provide the atom number statistics of the two crystal datasets** we used in our experiments as follows. As shown in the following table, crystal structures have more atoms than regular molecular graphs. Specifically, there are **more than 10k samples in The Materials Project with more than 50 atoms in a single cell**. Thus, larger complexity will be introduced when adding angular information compared with molecular graphs.
>
>      | Dataset | Mean atom numbers in a single cell | Max atom numbers in a single cell | Number of crystals with > 50 atoms in a single cel | Number of crystals with > 100 atoms in a single cell |
>      |----|----|----|----|----|
>      |JARVIS|10.1|140|308|3|
>      |The Materials Project|29.9|296|11911|1866|
>
> - Beyond this, following your suggestion, we **conducted ablation studies** to show the **high complexity of introducing angular information** to our Matformer in Sec 6.3 and in the table below. We introduced angular information using line graphs following ALIGNN, and added two edge layers of the original Matformer to deal with extra angular information. We use cosine plus RBF (as done by ALIGNN) and SBF kernels to encode angular information. The corresponding running times and final test MAE are shown in the following Table. It can be seen that introducing angular information largely increases the training time by nearly three times, and the performance gain is neglectable. It also shows that our proposed Matformer has great power for periodic graph learning, and is much more efficient.
>
>      |JARVIS Formation Energy|MAE|Time Per Epoch|Total Training Time|
>      |----|----|----|----|
>      |Matformer|0.0325|64 s|8.9 h|
>      |Matformer + Angle SBF|0.0332|173 s|23.8 h|
>      |Matformer + Angle RBF|0.0325|165 s|22.9 h|
>
> - However, we do believe introducing angular information in periodic graphs properly (without breaking periodic invariance) and efficiently (without introducing large complexity) is challenging yet promising, and we aim to tackle it in future research.
>
>
> > 8. Some relevant works [4, 5] regarding periodic graphs generation can be discussed in the related works section.
>
> Thank you for your suggestion. Following your suggestion, we added the discussion of CDVAE in the introduction section and Deep Generative Model for Periodic Graphs in the related work section in the revised paper.
>
> > 9. The code is not published for now to reproduce the results.
>
> Thank you for your suggestion and we attached our code in the supplementary materials.
>
> For limitations:
> > 1. The limitations of the model are not mentioned, the negative societal impacts of your work are marked as N/A.
>
> Thank you for letting us know your concerns.
>
> Following your suggestion, we added the limitations and the negative societal impacts in the final section and accordingly updated the Checklist in the revised paper.
>
>
> Hope we have addressed your concerns well, and we are looking forward to your reply. Thanks!

---

> ### Author Response · Authors · 2022-08-02
> **Revised paper according to your suggestions; necessarity of periodic invariance for valid crystal representations. (2nd Part)**
>
> > 2. It seems that section 3.1 and 3.2 can be moved to section 2 or they can just serve as a single section, since it seems that they are the motivation/theoretical ground of the proposed model not the model itself.
>
> Thank you for your suggestion.
>
> - Firstly, in the original version, we formatted sections 3.1 and 3.2 as separate sections rather than moving them to section 2. This was due to periodic invariance and periodic repeating patterns for periodic graph learning are often ignored and not addressed by previous works for periodic graphs. Specifically, periodic invariance is of great importance for periodic structures like crystals, but is rarely noticed by previous works. Treating atoms as individual nodes with corresponding 3D coordinates (like in molecular graphs) breaks periodic invariance. However, this strategy is widely used when handling periodic structures, even in some powerful methods such as Graphormer. Breaking periodic invariance will result in different representations for the same crystal structure, which could confuse machine learning models and produce suboptimal performance. Besides, when dealing with periodic crystals, no previous work tried to encode periodic patterns. Without encoding periodic patterns, the graph representation only captures a local structure, but fails to capture how the local structure expands itself in 3D space. Such an important component is even not noticed, not to mention solved in existing works. We propose to formally define them and conducted analysis on them.
>
> - However, after careful consideration, we do believe your suggestion is helpful for clarity.
> Thus, we followed your suggestion and formatted sections 3.1 and 3.2 to the new separate section 3 titled ‘Periodic invariance and periodic pattern encoding for crystals’ in the revised paper.
>
>
> > 3. Line 45, the paragraph is titled as "Crystal property prediction" but it seems that the whole paragraph is about the definition of periodic lattice. The title might be modified accordingly.
>
> Thank you for your suggestion.
>
> We followed your suggestion and edited the name from "Crystal property prediction" to "Crystal property prediction and crystal structures".
>
> > 4. It seems that E(3) invariance has been mentioned even in the abstract but it is formally defined in section 3.1.
>
> Thank you for bringing this question up.
>
> We followed your suggestion and added the description for E(3) invariance in the first paragraph in Section 1.
>
> > 5. Line 87, it mentions that "the structure f a cell remains the same when ...". Does the structure here mean the the output of the "f" function mentioned in the Definition 1 above?
>
> Thank you for bringing this up.
>
> The structure of a cell means the geometric 3D structure (conformation) of the unit cell. It is like the geometric structure of a molecule will not change when you rotate it in the 3D space. We also edited this line in section 3.1 of the revised paper to make it clearer.
>
> > 6. Line 90, It says "periodic invariance is also shown necessary for generating valid crystal representation". Why?
>
> Thank you for bringing this question up.
>
> - Firstly, as we mentioned in the original paper after this sentence, we show that **ignoring periodic invariance will end up with different representations for the same crystal structure, which will definitely generate unsatisfactory prediction results.**
>
> - For valid crystal representations, we need to map the same crystal to the same representation. As shown in Figure 2 in the original paper, we can actually have various unit cell structures for the same crystal by shifting the artificial periodic boundaries. **For these various unit cell structures of the same crystal, we need to map them to a unique representation. Hence, periodic invariance is also shown necessary for generating valid crystal representation.**
>
> If you have any follow up questions, we are more than glad to answer.
>
> We continue at the 3rd part as below:

---

> ### Author Response · Authors · 2022-08-02
> **Importance of periodic invariance clarified; motivation clarified.  (1st Part)**
>
> Thanks for your positive feedback on the importance of the task we focus on, the effectiveness of our proposed method. For the concerns and weaknesses, we provide a point-wise response as follows. We also revised the paper accordingly.
>
> For weaknesses:
> > 1. The motivation of the paper can be further improved: a) Why do we need to construct a graph from the unit-cell coordinates rather than directly using coordinates to conduct representation learning? What's the benefit to borrow the graph structure here? I believe there are a few works for representation learning on 3D positions such as [1], [2] and [3] b) What is the motivation to form self-connecting edges to contain the information of lattice matrix? Why not just concatenate it with the representation of a single unit cell?
>
> We respectfully disagree with this point.
>
> - For real-world prediction tasks, we usually need to satisfy invariance or equivariance. The concept of invairance lies in a scenario where when we rotate an object (like molecule), the coordinates for each atom in the molecule would change, but the molecule is still the same one!
> ML models should not treat the same molecule before and after rotation as different ones. Thus, we need to retain invariance in ML models. That’s why we introduce Unit Cell E(3) Invariance first at the beginning of Sec 3.1 in the paper. Directly taking coordinates may satisfy E(3) equivariance with delicate equivariant operations, but would not satisfy E(3) Invariance. For sure, it would not satisfy periodic invariance. Overall, **for (a), the reason that we need to construct a graph from the unit-cell coordinates and lattices rather than directly using coordinates to conduct representation learning is to ensure the periodic invariance property of periodic graphs.** Specifically, as shown in Figure. 2 in the main paper, we can get totally different 3D point clouds for the same periodic structure by shifting the artificial periodic boundaries. Because of this, traditional representation learning methods for molecular graphs and finite point clouds can not be used for crystals with periodic structures directly.  **Additionally, We also show that when coordinates are directly used (Graphormer), the periodic invariance is broken and performance drops significantly (OCgraph in Table. 5).**
>
> - For (b), Firstly, **if doing so, it is like we break the infinite periodic structure into a local unit cell structure and repeating patterns, then learn two representations and pass messages separately.** In this way, when performing message passing for every single atom in the unit cell, **the self repeating patterns in the infinite 3D space of that atom itself is not considered** (because the self-connecting edges are not modeled in the graph). Instead, **if the periodic patterns are encoded in self-connecting edges, when performing message passing, the self repeating patterns for a given atom is fully captured.**  Secondly, if we concatenate the lattice information with the representation of a single unit cell, it means we need a separate encoder for learning the lattice. Instaed, we can include the periodic patterns by simply adding self-connecting edges and no extra encoder is needed, which is more elegant.
>
> We continue at the 2nd part as below:

---

> ### Comment · Reviewer_oomT · 2022-08-05
> **I'm very satisfied with the responsed from the author**
>
> Thank you the author for the comprehensive response to my question. All my concerns have been addressed by the author so that I would like to support the publication of this article.

---

> > ### Author Response · Authors · 2022-08-08
> > **Dear Reviewer oomT**
> >
> > Thank you very much for your precious time! We are glad we addressed all your concerns. Your valuable comments and suggestions help improve our work a lot.
> >
> > Sincerely,
> >
> > Authors

---

### Official Review · Reviewer_AKt8 · 2022-07-11

**Rating:** 5
**Confidence:** 4
**Soundness:** 3 good
**Presentation:** 4 excellent
**Contribution:** 3 good

**Summary:**

This paper considers the property prediction tasks for crystal materials. This problem can be viewed as predicting a global target for a graph. The uniqueness is that the graph is periodic. The graph crystal structure repeats in the 3D space. Past works didn't really emphasize on this periodic invariance, or handle the periodic pattern encoding. The most commonly seen strategy is to set a fixed radius for an atom, and top $t$ closest neighbor atoms to this atom are connected to this atom with an edge. Such design implicitly considers the periodicity of the crytals. However, in some corner cases, such invariance may not be guaranteed. To this end, this paper proposes to explicitly capture the periodicity through self-connecting edges. Intuitively, the self-connecting edges connect one atom with its nearby duplicates. Since it is in the 3D space, there are multiple duplicates in different directions. So, multiple self-connecting edges are added to model the periodicity. The resulting model, Matformer, outperforms other GNN-based methods on two datasets, materials project and JARVIS.

**Questions:**

1. I belielive $\ell_1, \ell_2, \ell_3$ are 3d vectors with only one dim non-zero?
2. It's better to include $\ell$ as a superscript in your denoting $f^*_i$ for the $\ell$-th layer
3. What do you store in edge feature vector?
4. In Table 1, it seems that the numbers from your reproduction and the original paper differ by a lot. Why is that?
5. Why GATGNN etc. do not show up in Table 3?
6. Whether to add the angle information is always debatable in this field. ALIGNN often becomes the second best numbers and it encodes angles. But you claimed angles "induce high complexity", could you clarify on this?
7. For each material property, do you train a separate model? or you use one learnt representaion for all?

**Limitations:**

1. I do believe it would be better for authors to include the discussion for the comparison between Euclidean neural nets and graph neural nets in encoding the crystals.
2. Some interpretations on the results can help me to better understand the perforamnce gain.
3. More evaluation metrics would make it more convincing.

**Strengths And Weaknesses:**

Pros:
1. The paper is written clearly. Though there are some typos, the general idea and methods are easy to follow. And the real-world problem is also important in materials science.
2. The ablation study is presented. I think the key argument of this paper is the periodic invariance. So proving the importance of the periodic module is critical.
3. Multiple datasets are included for validation.

Cons:
1. Essentially, the paper extends the existing graph based methods for crystal property prediction. However, another direction is Euclidean neural networks. These models are not compared in this paper, such as E3NN.
2. The authors decide to drop the angle information due to high complexity. But ablation study should be provided to support this claim.
3. As I mentioned, the ablation study is critical for performance and operations. It would be better to expand the ablation study section and move the dataset descriptions to appendix.
4. From the comparison numbers, the improvements over others like ALIGNN are not that significant.
5. Experiments only evaluate MAE. More metrics can be added.

---

> ### Author Response · Authors · 2022-08-02
> **Limitations addressed(4th Part)**
>
> For limitations:
>
> > 1. I do believe it would be better for authors to include the discussion for the comparison between Euclidean neural nets and graph neural nets in encoding the crystals.
>
> Thank you for the suggestion.
>
> - We **added the discussion for the comparison between Euclidean neural nets (Nequip) and our Matformer in the related work section in the revision.**
>
> - Beyond this, we also **reimplemented Nequip for JARVIS five tasks**, and the results are shown in above answers for Cons 1. We can add the results in Table 2 in the paper if you suggest so.
>
> > 2. Some interpretations on the results can help me to better understand the performance gain.
>
> Thank you for bringing this up.
>
> We **added the EwT and wT comparisons with previous SOTA ALIGNN** as described in answers for Cons 4. **Overall, it can be seen from metrics of EwT, wT, and MAE that our Matformer outperforms ALIGNN consistently under different metrics.** As discussed in details in Cons 4, **better EwT and wT means our methods can generate more predictions which are practically useful.**
>
> > 3. More evaluation metrics would make it more convincing.
>
> Thank you for bringing this up.
>
> We **added EwT as the evaluation metric** in the main paper in Section 6.2, and **more results of EwT and wT** are shown in the answers for Cons 4.
>
> Hope we have addressed your concerns well, and we are looking forward to your reply. Thanks!

---

> > ### Comment · Reviewer_AKt8 · 2022-08-05
> > **Some further questions and clarifications**
> >
> > > We noticed that E3NN is a newly released paper in July 2022
> >
> > I'm not sure which paper you are referring to. But here are some papers in 2021 applying E3NN to materials prediction tasks.
> >
> > Chen, Zhantao, et al. "Direct prediction of phonon density of states with Euclidean neural networks." Advanced Science 8.12 (2021): 2004214.
> >
> > Batzner, Simon, et al. "SE (3)-Equivariant Graph Neural Networks for Data-Efficient and Accurate Interatomic Potentials." arXiv preprint arXiv:2101.03164 (2021).
> >
> > [E3NN](https://e3nn.org/) codebase was built back in 2020.
> >
> > Tasks may be different. But there is no doubt these papers including yours all focus on graph encoding.
> >
> > > From the comparison numbers, the improvements over others like ALIGNN are not that significant.
> >
> > 1. When I mentioned this, I was actually looking at Table 1 and Table 2. MAE, as you explained, is most widely adopted in this field. For instance, in Table 1, formation energy 0.022 -> 0.021, band gap 0.218 -> 0.211; in Table 2, 0.0331 -> 0.0325 etc. I appreciate that Matformer consistently outperform other baselines like ALIGNN, but I'm not sure how much such small gaps make sense to materials scientists.
> >
> > 2. Also, it looks like you didn't mention the model sizes (correct me if I'm wrong). If the model sizes are quite different, comparing their numbers are unfair. This often happens to transformer-based models since the size can go high very easily.
> >
> > > comments on the model architecture
> >
> > In general, Matformer has not jumped out of the framework of a series GNNs like GATGNN. The main contribution is the self-loop. While I appreciate the consistent improvements, the gains are not as large as the authors implied in previous sections (at least on MAE). I think for a NeurIPS paper, I'm expecting slightly more.
> >
> > &nbsp;
> >
> > With these being said, I appreciate authors' effort in considering my suggestions and providing additional numbers. I'm willing to raise my eval rating a bit.

---

> > > ### Author Response · Authors · 2022-08-06
> > > **Answers for main contributions, performance gains and novelties (3rd Part)**
> > >
> > > > 5. The main contribution is the self-loop.
> > >
> > > The usage of self-loops is indeed the final approach we used to encode periodic patterns without breaking periodic invariance. But we kindly disagree that it is our main contribution, as we do NOT randomly add self-loops. Our technical approaches are well motivated (precise crystal graph construction), problems are well defined and analyzed (periodic invariance and periodic pattern encoding),  and technical approaches are provable. As a result, we propose two techniques to solve the periodic invariance, and *propose to add self-loops to solve periodic patterns encoding*.
> > >
> > > - Our work is firstly motivated by precise graph construction for periodic crystal structures. To this end, we first proposed to define periodic invariance and periodic pattern encoding for periodic graphs, which are rarely noticed by the community and of great importance for the representation learning of periodic graphs like crystals. We then give detailed analysis of why these two components are important in Section 3. Beyond this, we propose two solutions to preserve periodic invariance with corresponding proofs in Appendix A.2. These two methods can be used safely by the community and future works without worrying about breaking periodic invaraince. We then propose a new, elegant, and efficient way to encode periodic patterns when constructing graphs. Usage of self-loop is the technical approach to encoding periodic patterns while preserving periodic invariance.
> > > - Also, the proposed Matformer is demonstrated to be **powerful** (outperforms previous SOTA consistently) and **efficient** (nearly 3 times faster in training and inference time). Due to the nature of period graphs, Matformer is very different from transformers for texts, images, and regular graphs. the Matformer architecture is also different from the GATGNN architecture. All the proposed technical solutions in this work are new, effective, and efficient.
> > > - Additionally, **we notice the comparisons between previous works are not fair and honest.** They compare with each other with different random split seeds and different numbers of training samples. **We are the first work to fairly benchmark all these baseline methods on The Materials Project and JARVIS datasets. We believe the fair benchmarking of previous methods provided by our work is of great value for the community, and can generate immediate impacts for ML on crystals**.
> > > - Overall, we think the novelty is sufficient due to 1). we discover, formally define, and propose approaches to resolve two vital components for crystal graph construction; 2). we provide a novel, powerful, and efficient crystal learning model; and 3). we are the first work to compare all baselines fairly with exactly the same data splits.
> > >
> > > > 6. While I appreciate the consistent improvements, the gains are not as large as the authors implied in previous sections (at least on MAE). I think for a NeurIPS paper, I'm expecting slightly more.
> > >
> > > We respectfully disagree with this.
> > >
> > > - We show the large performance gains **in MAE** of our methods beyond ALIGNN as described in the above answer for question 2 (**7.45% and 5.94%**).
> > > - We show the significant performance gains **in EwT/wT** of our methods beyond ALIGNN as described in the above answer for question 2. (**24.69% in terms of EwT and wT 0.01 for JARVIS, 11.78% in terms of EwT and wT 0.02 for JARVIS, 24.17% in terms of EwT and wT 0.01 for MP, 11.16% in terms of EwT and wT 0.02 for MP**)
> > > - We illustrate the novelties and contributions of our work in the above answer for question 5.
> > >
> > > Hope we have addressed your concerns well, and we are looking forward to your reply. Thanks!

---

> > > > ### Comment · Reviewer_AKt8 · 2022-08-07
> > > > **Further reviewer replies**
> > > >
> > > > > For The Materials Project, ..., the average performance gain is 7.45%. For the JARVIS dataset, the average performance gain is 5.94%.
> > > >
> > > > First, in general, I think it is more appropriate to view each task individually. For instance, if you have x% improvement on imagenet and y% improvement on coco, one typically doesn't say (x+y)/2% improvement on average. But, of couse, if you show the average, I could understand what you mean. Second, the averages are really brought by the improvements on Bulk Moduli prediction task (15.7%, 15.8%). If you drop this one, the average of others would immediately drop below 5%. In many ML tasks, around 3-4% gains are generally tunable.
> > > >
> > > > > we included two metrics EwT and wT
> > > >
> > > > First, the concepts of EwT and wT are newly introduced in the rebuttal period, which were not mentioned in the initial version. Second, let's say it's ok to introduce new concepts in the revision during rebuttal. EwT and wT, as you mentioned, were introduced in an open challenge in 2020. Prior works you followed like ALIGNN came out after the competition, but it seems that those works didn't really consider EwT in their papers. I'm not material scientist and you are submitting to a CS conference, I think I can be a bit conservative on the new metrics you introduced in the revision. Given that you said most prior works adopt MAE, I think it should be ok for me to value MAE results more. If you really want to advocate for such new metrics, I think the paper could be better shaped and why these new metrics are significant should be elaborated. Third, the new metrics were introduced in another dataset OC20, but you didn't really try your method on OC20 dataset. On the other hand, materials project and jarvis datasets which you indeed evaluated didn't mention such metrics previously. So, it would look a bit inconsistent here.
> > > >
> > > > > But we kindly disagree that it is our main contribution, as we do NOT randomly add self-loops
> > > >
> > > > First, I think NO one would add self-loop randomly. In fact, Technically, previous works [1, 2] and many other works (if you search for related papers) have already introduced self-loops, studies or tested self-loops for different purposes or from different perspectives. Second, I understand you are adding self-loops motivated by a novel real-world application (crystal structure) and Matformer is a novel variant of conventional GAT. So I more see it a novel adapted application of GAT to materials property prediction from an ML perspective. I would be more convinced if the same model can be applied to other types of periodic graphs as well. Meanwhile, the experiments and rebuttals look solid and thus I rate the paper positive.
> > > >
> > > >
> > > > [1] Hamilton, William L. "Graph representation learning." Synthesis Lectures on Artifical Intelligence and Machine Learning 14.3 (2020): 1-159.
> > > >
> > > > [2] Wu, Felix, et al. "Simplifying graph convolutional networks." International conference on machine learning. PMLR, 2019.

---

> > > > > ### Author Response · Authors · 2022-08-08
> > > > > **Response for Further replies: contributions and differences with GATs (2nd Part)**
> > > > >
> > > > > > 6. About contributions and differences with GATs
> > > > >
> > > > > - we believe **model architecture design of Matformer is only one of our several contributions**. For the model architecture, we added a detailed comparison with GATGNN and GAT in the above answer in the second round for question 4 and in our revised paper in the related work section. **There are limited similarities between Matformer and GATs, and the major difference lies in that, GATs do NOT use self-attention [1], while Matformer is based on self-attention.** Self-attention is currently the key component of almost all transformers. If anything is unclear, we are more than glad to make it clearer and will respond as quickly as possible.
> > > > >
> > > > > - More importantly, **we are the first to formally notify and define two important components for crystal graph construction: periodic invariance and periodic patterns encoding.** Specifically, periodic invariance is of great importance for periodic structures like crystals but is rarely noticed by previous works. Treating atoms as individual nodes (like in molecular graphs) breaks periodic invariance. However, this strategy is widely used when handling periodic structures, even in some powerful methods such as Graphormer. Breaking periodic invariance will result in different representations for the same crystal structure, which could confuse machine learning models and produce suboptimal performance. Besides, when dealing with periodic crystals, no previous work tried to encode periodic patterns. Without encoding periodic patterns, the graph representation only captures a local unit (local structure) but fails to capture how the unit (local structure) expands itself in 3D space. Such an important component is even not noticed, not to mention solved in existing works. We also design several effective and efficient techniques to fulfill these two components. To this end, **we believe the discovery, formal definition, and novel technical solutions of periodic invariance and periodic patterns encoding are of great value, importance, and significance to the community.**
> > > > >
> > > > > - Additionally, **we notice the comparisons between previous works are not fair and honest. We are the first work to fairly benchmark all these baseline methods on The Materials Project and JARVIS datasets.** **We believe the fair benchmarking of previous methods provided by our work is of great value for the community and can generate immediate impacts for ML on crystals.**
> > > > >
> > > > > Thank you very much for all your suggestions and discussions, which help improve our work a lot! If you have any other concerns, please let us know and we are more than glad to answer.
> > > > >
> > > > > > Reference
> > > > >
> > > > > [1] Vaswani, Ashish, et al. "Attention is all you need." Advances in neural information processing systems 30 (2017).

---

> > > > > ### Author Response · Authors · 2022-08-08
> > > > > **Response for Further replies: clarifications (1st Part)**
> > > > >
> > > > > We genuinely thank you for the recognition of our solid experiments and rebuttals. We sincerely appreciate all the suggestions, the recognition, and the positive rating from you. We provide a point-wise response as follows.
> > > > >
> > > > > > 1. First, in general, I think it is more appropriate to view each task individually.
> > > > >
> > > > > - We agree with your thought.
> > > > > - We understand you may be concerned with the performance gains for some tasks, like formation energy and band gap for these two datasets (your examples in the last round of comments were for these two tasks).
> > > > > - For Formation Energy in The Materials Project and JARVIS, the performance gains beyond ALIGNN are **4.5% and 1.8%** in terms of MAE and **11.9% and 4.0%** in terms of EwT 0.01. These results show that our Matformer is **more powerful** than ALIGNN. Additionally, energy predictions within a small threshold are **meaningful to real-world applications** and to **the material science community.** Beyond this, we can see that when more training samples are available, the performance of our Matformer increases a lot larger compared with ALIGNN, which reveals the **huge potential of Matformer when larger datasets are available.**
> > > > > - For band gap (OPT) in JARVIS and band gap in The Materials Project, the performance gains beyond ALIGNN are **3.2% and 3.5%** in terms of MAE and **36.5% and 15.6%** in terms of EwT 0.01.
> > > > > - **We respectfully disagree that these kinds of improvements are tunable for these tasks** and we believe the performance gains are significant.
> > > > >
> > > > > > 2. First, the concepts of EwT and wT are newly introduced in the rebuttal period, which was not mentioned in the initial version. Second, let's say it's ok to introduce new concepts in the revision during rebuttal. EwT and wT, as you mentioned, were introduced in an open challenge in 2020. Prior works you followed like ALIGNN came out after the competition, but it seems that those works didn't really consider EwT in their papers. I'm not a material scientist and you are submitting to a CS conference, I think I can be a bit conservative on the new metrics you introduced in the revision.
> > > > >
> > > > > - Thank you very much for bringing this up.
> > > > > - We **genuinely followed your suggestions in the initial Cons 4, 5 and limitations 2, 3 to include several different evaluation metrics to show the performance gains.** **We deeply appreciated all your suggestions and understand your concerns.** As a result, we added comparisons with ALIGNN in terms of EwT and wT that are different from MAE in the first round of rebuttal.
> > > > > - **You are right that previous works didn't really consider EwT in their papers because they were all proposed before the OC20 data and competition (the year 2020) where EwT and wT were introduced, and the only work after the competition is ALIGNN, where EwT was not used either. But this does not mean that EwT and wT are not meaningful in material science.** We believe EwT and wT are meaningful metrics because they can tell a lot about how many predictions can be used practically.
> > > > >
> > > > > > 3. Given that you said most prior works adopt MAE, I think it should be ok for me to value MAE results more. If you really want to advocate for such new metrics, I think the paper could be better shaped and why these new metrics are significant should be elaborated on. On the other hand, the materials project and jarvis datasets which you indeed evaluated didn't mention such metrics previously. So, it would look a bit inconsistent here.
> > > > >
> > > > > - Thank you very much for your suggestions.
> > > > > - As you may notice, **we did add the significance analysis of EwT metrics in Section 6.2, in lines 306-309 in the revised paper** in the first round of rebuttal.
> > > > >
> > > > > > 4. Third, the new metrics were introduced in another dataset OC20, but you didn't really try your method on the OC20 dataset.
> > > > >
> > > > > - Thank you very much for your suggestions.
> > > > > - As you may notice, OC20 is a dataset for interactions between adsorbates and catalysts. Although catalysts can have periodic structures on the x and y-axis, adsorbates do not have periodic structures. So **the whole input structure is not periodic. Our Matformer is designed for periodic graphs like crystals and can not be applied directly to OC20.** Hence, OC20 is not used to evaluate performance.
> > > > >
> > > > > > 5. On the other hand, materials project and jarvis datasets which you indeed evaluated didn't mention such metrics previously. So, it would look a bit inconsistent here.
> > > > >
> > > > > - Thank you very much for bringing this up.
> > > > > - Yes, EwT and wT are not introduced in previous works. Previous works (except for ALIGNN) didn't consider EwT in their papers because they were all proposed before the OC20 data and competition (the year 2020) where EwT was introduced. The only work after the competition is ALIGNN, where EwT is not used either, actually. We added the significance analysis of EwT metrics in Section 6.2, in lines 306-309 in the revised paper in the first round of rebuttal.
> > > > >
> > > > > The 2nd part is as follows:

---

> > > ### Author Response · Authors · 2022-08-06
> > > **Provided model sizes comparison; clarified differences with GATGNN (2nd Part)**
> > >
> > > > 3. Also, it looks like you didn't mention the model sizes (correct me if I'm wrong). If the model sizes are quite different, comparing their numbers is unfair. This often happens to transformer-based models since the size can go high very easily.
> > >
> > > Thank you very much for your suggestion.
> > >
> > > - We added the comparison of model size between Matformer and ALIGNN in the revised paper (Table 3) and as shown below. It can be seen that compared with previous ALIGNN, our ***Matformer actually has the smaller model size and achieves better results consistently, in terms of not only MAE, but also EwT and wT.***
> > >      | Matformer | ALIGNN |
> > >      |----|----|
> > >      |11528236 (11.0 MB) |16164044 (15.4 MB and 40.2% more than Matformer)|
> > >
> > > > 4. For the model Architecture. In general, Matformer has not jumped out of the framework of a series GNNs like GATGNN.
> > >
> > > We respectfully disagree with this.
> > >
> > > - First of all, we think our major novelty lies in graph construction with periodic invariance and periodic pattern encoding. In terms of space, the formal definition, importance analysis, and technical solutions of these two components occupy 3 pages. **The graph construction in GATGNN fails to integrate periodic patterns, and the constructed graphs are not precise enough**. Regardless of this, even for the network architecture, we believe our Matformer is different from GATGNN.
> > >
> > > - (1) **Basic layers and attention operation are different**: Our Matformer belongs to the Transformer framework, which is different from the GAT[1] framework. GAT framework calculates attention weights using the concatenation of transformed source and target node features, followed by a **linear transformation** and softmax. While the Transformer framework calculates attention weights using the **Hadmard Product** between query and key vectors, followed by softmax or other operations. Actually, GAT computes a very limited kind of attention and the ranking of the attention scores is unconditioned on the query node, and the corresponding negative consequences are detailed in GATv2[2]. But the Transformer framework does not have this problem.
> > >
> > > - (2) **Softmax constraints the capability of GATGNN of distinguishing nodes with different degrees**: GATGNN uses softmax to aggregate information from neighbors for a given node $i$, but softmax limits the capability of GATGNN to distinguish nodes with different degrees. For example, consider two different inputs for the attention layer. One is that node $i$ which has 3 neighbors with the same pairwise distances d and node type t, the other is node $i$ which has 1 neighbor with pairwise distance d and node type t. The GATGNN will produce the same results for these two cases because all information from these 3 neighbors is the same. But our Matformer can distinguish these two cases by using layernorm and sigmoid instead of softmax, as we mentioned in Section 4.2. We also conducted experiments comparing the operation of softmax and layernorm with sigmoid in Appendix A.6, where we show our methods are more effective.
> > >
> > > We added the discussion with GATGNN in the related work section in the revised paper.
> > >
> > > > References
> > >
> > > [1] Veličković, Petar, et al. "Graph Attention Networks." International Conference on Learning Representations. 2018.
> > >
> > > [2] Brody, Shaked, Uri Alon, and Eran Yahav. "How Attentive are Graph Attention Networks?." International Conference on Learning Representations. 2021.
> > >
> > > We continue the 3rd part as below:

---

> > > ### Author Response · Authors · 2022-08-06
> > > **Included related works; demonstrated large performance gains, huge potential, and significance of Matformer beyond ALIGNN (1st Part)**
> > >
> > > Thank you very much for the following questions and clarifications.
> > >
> > > > 1. I'm not sure which paper you are referring to. But here are some papers in 2021 applying E3NN to materials prediction tasks. Chen, Zhantao, et al. "Direct prediction of phonon density of states with Euclidean neural networks.". Batzner, Simon, et al. "SE (3)-Equivariant Graph Neural Networks for Data-Efficient and Accurate Interatomic Potentials." E3NN codebase was built back in 2020. Tasks may be different. But there is no doubt these papers including yours all focus on graph encoding.
> > >
> > > - Sorry for our misunderstanding. We thought the referred paper was *Mario Geiger, and Tess Smidt, e3nn: Euclidean Neural Networks, arXiv preprint arXiv:2207.09453 (2022)*, because the title of this paper contains the term *E3NN*. Now we are clear that you were talking about the general E(3) neural networks, and thank you for your clarifications! Previously, we added the *Nequip* paper as you mentioned. This time we also included and discussed the paper *Zhantao Chen et al, Direct prediction of phonon density of states with Euclidean neural networks, Advanced Science 8.12 (2021)* in line 31 in our paper. We did some literature review and believe that we should have included all recent works about e3nn on materials.
> > >
> > > - In our first round of responses (answers for Cons 1), we added discussions and comprehensive experiments about *Nequip*, which is the model you mentioned as *Simon Batzner, et al,SE (3)-Equivariant Graph Neural Networks for Data-Efficient and Accurate Interatomic Potentials, arXiv preprint arXiv:2101.03164 (2021)*. As *Nequip* applies E3NN to materials prediction tasks, we reimplemented and retrained it on JARVIS five tasks, and the results were included in In our first round of responses (answers for Cons 1). The performance of *Nequip* is much worse due to the lack of periodic pattern encoding. As we also mentioned, those results can be added in the paper if you suggest so.
> > >
> > > > 2. When I mentioned this, I was actually looking at Table 1 and Table 2. MAE, as you explained, is the most widely adopted in this field. For instance, in Table 1, formation energy 0.022 -> 0.021, band gap 0.218 -> 0.211; in Table 2, 0.0331 -> 0.0325 etc. I appreciate that Matformer consistently outperforms other baselines like ALIGNN, but I'm not sure how much such small gaps make sense to materials scientists.
> > >
> > > Thank you for bringing this up.
> > >
> > > - Yes, in terms of MAE, the performance gains of our methods beyond ALIGNN are not as large as those of ALIGNN beyond other baseline methods. In this paper, we tend to design methods that are both powerful and efficient. Our methods are three times faster than ALIGNN, which is very important in practice.
> > >
> > > - For The Materials Project, in terms of MAE, the performance gains beyond ALIGNN are 4.5%, 3.2%, 15.7%, 6.4%, and **the average performance gain is 7.45%**. For the JARVIS dataset, the performance gains beyond ALIGNN are 1.8%, 3.5%, 5.4%, 15.8%, 3.2%, and **the average performance gain is 5.94%**. Hence, we believe the performance gains beyond ALIGNN in terms of MAE are still significant.
> > >
> > > - To demonstrate the significance to materials scientists, as described in our answers for Cons 4 and Cons 5 and limitations 2 in the first round, we included two metrics EwT and wT. These two metrics can demonstrate how many predictions from models thus can be practically useful. From the comparison results, we can see that our Matformer consistently outperforms ALIGNN with significant margins, in terms of EwT 0.01, EwT 0.02, wT 0.01, and wT 0.02 for all these tasks in The Materials Project and JARVIS. As we mentioned, we have put most of those results in the revised paper.
> > >
> > > - Additionally, the performance gains of our Matformer beyond ALIGNN in terms of EwT mainly come from more accurate energy predictions within an absolute error of 0.01 (compared with 0.02). The demonstration on formation energy is particularly shown in the following table, and more results were already provided in Cons. 4 in the last round. This indicates that Matformer generates more accurate predictions when the prefixed error (threshold) is more strict. Compared with JARVIS, the Materials Project has 15422 more training samples. With more training samples available, **the percentage of predicted energies obtained by Matformer within 0.01 increases by 14.69% (55.86%-41.17%)**, ***which is significantly better than ALIGNN***, **revealing the huge potential of Matformer when larger crystal dataset is available**. We believe these results **make sense for material scientists** because of **much more practically useful energy predictions**, and **reliability and huge potentials when larger datasets are available** of Matformer.
> > >
> > >      | Formation Energy JARVIS | EwT 0.01 | EwT 0.02 | Formation Energy MP | EwT 0.01 | EwT 0.02 |
> > >      |----|----|----|----|----|----|
> > >      |ALIGNN|39.59%|59.64%||49.94%|71.10%|
> > >      |Matformer|41.17%|60.25%||55.86%|75.02%|

---

> ### Author Response · Authors · 2022-08-02
> **Added EwT and wT as new evaluation metrics; questions answered and clarified. (3rd Part)**
>
> > 5. Experiments only evaluate MAE. More metrics can be added.
>
> Thank you for bringing this up.
>
> - For the evaluation metrics, we directly follow previous works (CGCNN, MEGNET, SchNet, ALIGNN, etc. ) for crystal property prediction and use MAE as evaluation metric.
>
> - Following your suggestion, we revised the paper and added energy within threshold (EwT) as an evaluation metric in Section 6.2. `More results of prediction within threshold as provided above can be added if you suggest so.` We think the metrics EwT and wT are totally different from MAE (also RMSE). EwT and wT evaluate the percentage of precise predictions within a tight threshold, and can be used to demonstrate the percentage of useful predictions of a prediction model for crystals. These two metrics are pretty useful in practice. MAE and RMSE are similar, and they both somehow evaluate the divergences in distributions between the whole predictions and ground truth. Hence, we do believe EwT is better due to a new perspective, and we may add results on RMSE in the next version.
>
> For Questions:
>
> > 1. I belielive $l_1, l_2, l_3$ are 3d vectors with only one dim non-zero?
>
> - As we mentioned in the original version in Section 2, crystals usually possess irregular shapes in practice and $l_1, l_2, l_3$ are not always orthogonal in 3D space.
>
> - Hence, each of $l_1, l_2, l_3$ is indeed a 3d vector. However, for each of them, all the three entries can be non-zero, because the lattice structure is not always cubic.
>
> > 2. It's better to include as a superscript in your denoting for the l-th layer
>
> Thank you for your suggestion, we edited it in the revision in Section 4.2.
>
> > 3. What do you store in edge feature vector?
>
> As we mentioned in Section 2 (line 70) in the original version and line 69 in the revised version, we use Euclidean distance as the initial edge feature. Also, in our original version, we mentioned in the Appendix Matformer Configurations section that we map the Euclidean distance to a 128-dimensional embedding using 128 RBF kernels with centers from 0.0 to 8.0.
>
> > 4. In Table 1, it seems that the numbers from your reproduction and the original paper differ by a lot. Why is that?
>
> - Firstly, as stated in the original paper in the experimental results section (Sec 6.2) for The Materials Project, we noticed **previous works actually compared with each other with different random splits and even different numbers of training samples.** We think **these comparisons are not fair and honest enough.** For example, CGCNN only uses 28046 training samples for the task of Formation Energy in the Materials Projects, and the original MAE result is 0.039. To make the comparison between all baselines fair, we retrained them using exactly the same data split and also tuned some hyperparameters including learning rates and optimizer choices, to obtain the best results for each baseline model.
>
> - Beyond this, **our retrained results for baseline methods are better than their reported results in most cases.** We think the gaps are partially due to different data splits as stated above.
>
>
> > 5. Why GATGNN etc. do not show up in Table 3?
>
> - Firstly, for the JARVIS dataset, we directly follow the experimental settings of ALIGNN, and baselines including GATGNN are not included in ALIGNN settings in their paper.
>
> - However, following your suggestion, we **retrained and reimplemented GATGNN, MEGNET and SchNet for these five tasks in JARVIS.** We revised the paper and reported those results in Table 2.
>
> > 6. Whether to add the angle information is always debatable in this field. ALIGNN often becomes the second best numbers and it encodes angles. But you claimed angles "induce high complexity", could you clarify on this?
>
> Thank you for bringing this question up.
>
> - Firstly, as analyzed in section 6.3 and Appendix A.6 in the revised versions, adapting angular information introduces large complexity. **We show that for the original crystal graph with n nodes and 6n edges, the corresponding line graph with angles will have 6n nodes and 66n edges, leading to high computational cost.**
>
> - Also, following your suggestion, we conducted **running time experiments when angular information is included**, the details can be found in the above answers for Cons. 2. We also revised the paper and added those results along with analysis in ablation study in Section 6.3 and Appendix A.6.
>
>
> > 7. For each material property, do you train a separate model? or you use one learnt representation for all?
>
> Thank you for bringing this question up.
>
> - As we described in Section 5.1 in the original version and section 6.1 in the revised version, we slightly adjust learning rates and training epochs for different tasks, and we provide detailed hyperparameter configurations for different tasks in Appendix Matformer Configurations.
>
> - So yes, **following previous works (they did the same)**, we train a separate model for each material property.
>
> We continue at the 4th part as below:

---

> ### Author Response · Authors · 2022-08-02
> **Paper revised according to suggestions; EwT and wT added as new evaluation metrics to reveal Matformer’s great power from new perspectives. (2nd Part)**
>
> > 3. As I mentioned, the ablation study is critical for performance and operations. It would be better to expand the ablation study section and move the dataset descriptions to appendix.
>
> Thank you for your suggestion. We agree that the ablation study is critical for performance and operations. we moved the dataset descriptions to Appendix. We also added the ablation studies about angular information in Section 6.3 and Appendix A.6, as discussed above.
>
> > 4. From the comparison numbers, the improvements over others like ALIGNN are not that significant.
>
> We respectfully disagree with this.
>
> `Firstly,` compared with previous SOTA ALIGNN, our Matformer consistently outperforms ALIGNN on two widely used material dataset for 9 different tasks in MAE, and is three times faster in total training time and nearly three times faster in reference time.
>
> `We also added two new metrics`, including energy within threshold (EwT) and prediction within threshold (wT) for evaluation. These metrics are from paper [1], and are **well recognized by the community.** This is because they provide a different perspective from MAE (or RMSE), and are meaningful in practice. For example, EwT measures the percentage of estimated energies that are likely to be practically useful when the absolute error is within a certain threshold, and this threshold could be 0.01 or 0.02.
> We then conducted comparison with ALIGNN in terms of EwT and wT to better demonstrate the performance gains.
>
> **For all the tasks in JARVIS and formation energy and bandgap tasks in the Materials Project, we use the official prediction results for test sets from ALIGNN to calculate corresponding results. The bulk moduli and shear moduli are not included because no official results from ALIGNN are released.** We use EwT 0.02 to denote energy prediction error within threshold of 0.02 and  EwT 0.01 to denote energy prediction error within threshold of 0.01 for tasks of formation energy and total energy. We use wT 0.02 to denote prediction error within threshold of 0.02 and  wT 0.01 to denote prediction error within threshold of 0.01 for other tasks. The results are shown in the following tables.
>
> 1). Formation Energy for JARVIS and the Materials Project
>
> | JARVIS | EwT 0.01 | EwT 0.02 | MP | EwT 0.01 | EwT 0.02 |
> |----|----|----|----|----|----|
> |ALIGNN|39.59%|59.64%||49.94%|71.10%|
> |Matformer|41.17%|60.25%||55.86%|75.02%|
>
> It can be seen from this table that Matformer has more accurate predictions for crystals. Interestingly, the performance gains of our Matformer beyond ALIGNN in terms of EwT mainly **come from more accurate energy predictions within an absolute error of 0.01.** This indicates that Matformer generates more accurate predictions when the prefixed error (threshold) is more strict. Compared with JARVIS, the Materials Project has 15422 more training samples. With more training samples available, the percentage of predicted energies obtained by Matformer within 0.01 increases by 14.69%, which is significantly better than ALIGNN, revealing **the huge potential of Matformer when larger crystal datasets are available.** We also revised Section 6.2 of the paper and added experiments and discussions on the new metric EwT.
>
> 2). Band Gap (MBJ) and Band Gap (OPT) for JARVIS
>
> | Band Gap (MBJ)  | wT 0.01 | wT 0.02 | Band Gap (OPT)  | wT 0.01 | wT 0.02 |
> |----|----|----|----|----|----|
> |ALIGNN|18.10%|31.87%||48.73%|61.79%|
> |Matformer|31.70%|43.64%||56.31%|64.02%|
>
> It can be seen from the table that there are 13.6% gain in terms of wT 0.01 for Band Gap (MBJ) and 7.58% gain in terms of wT 0.01 for Band Gap (OPT), which are very significant.
>
> 3). Ehull and Total Energy for JARVIS
>
> | Ehull  | wT 0.01 | wT 0.02 | Total Energy  | EwT 0.01 | EwT 0.02 |
> |----|----|----|----|----|----|
> |ALIGNN|22.92%|39.52%||35.09%|55.20%|
> |Matformer|28.37%|44.81%||36.84%|57.36%|
>
> 4). Band Gap for The Materials Project
>
> | MP-BandGap | wT 0.01 | wT 0.02 |
> |----|----|----|
> |ALIGNN|19.79%|30.15%|
> |Matformer|27.01%|35.22%|
>
> `Overall,` **Matformer outperforms ALIGNN in terms of MAE, EwT 0.01, EwT 0.02, wT 0.01 and wT 0.02 for all these seven tasks in JARVIS and the Materials Project, revealing the significant performance gain and modeling capacity beyond previous state-of-the-art ALIGNN.** We included the comparison on formation energy-MP, formation energy-JARVIS and total-energy JARVIS in Section. 6.2 of the revised paper due to space limit. All the results can be added to the main paper if you suggest so. In addition, Matformer is 3 times faster than ALIGNN, thus is much more efficient.
>
> > Reference
>
> [1] Chanussot, Lowik, et al. "Open catalyst 2020 (OC20) dataset and community challenges." ACS Catalysis 11.10 (2021): 6059-6072.
>
> We continue at the 3rd part as below:

---

> ### Author Response · Authors · 2022-08-02
> **Included E3NN and Nequip; conducted experiments using Nequip official code on JARVIS five tasks; complexity analysis of adding angular information (1st Part)**
>
> Thanks for your positive feedback on the clarity of writing, the importance of our focused problem, and experiments including ablation study and evaluations on multiple datasets. For the concerns and weaknesses, we provide a point-wise response as follows. We also revised the paper accordingly.
>
> For Cons:
> > 1. Essentially, the paper extends the existing graph-based methods for crystal property prediction. However, another direction is Euclidean neural networks. These models are not compared in this paper, such as E3NN.
>
> Thank you for bringing this question up. We noticed that E3NN is a newly released paper in July 2022, which is after the Neurips submission deadline.
>
> - Firstly, following your suggestion, we **revised the paper and included the discussion** for the comparison with Euclidean neural nets including E3NN and Nequip in encoding crystal structures in introduction & related work sections.
>
> - Additionally, we also **reimplemented Nequip** (a Euclidean neural network and E(3) Equivariant Neural Network) based on the official code and conducted a comparison between Nequip and Matformer on the five tasks of JARVIS. The results are shown in the below table. The significant performance gain reveals the importance of the encoding of periodic patterns and the modeling capacity of the Matformer layer. `We can add those results in the paper if you suggest so.` And we will add experimental results of E3NN for comparison in the future.
>
> | JARVIS MAE | Formation energy | band gap (OPT) | Total Energy | Ehull | band gap (MBJ) |
> |----|----|----|----|----|----|
> |Nequip|0.048|0.26|0.048|0.17|0.50|
> |Matformer|0.0325|0.137|0.035|0.064|0.30|
>
> > 2. The authors decide to drop the angle information due to high complexity. But ablation study should be provided to support this claim.
>
> Thank you very much for the suggestion.
>
> - Firstly, following your suggestion, we **provided the complexity analysis** of introducing angular information in crystal graphs in section 6.3 and Appendix A.6. Assume that we have *n* atoms in a single cell and thus *n* nodes in the original graph. Also assume that every node has at least 12 neighbors (following the graph construction method of Matformer) and there are no self-connecting edges. This will result in a graph G=(V,E) where |V| = n and |E| = 12/2 n = 6n. When converting graph G into L(G), every edge is treated as a node in the line graph. So we have 6n nodes in the line graph. Every edge in the original graph is connecting 2 nodes and every node has (12 - 1) other edges, resulting in 22 neighboring edges for each edge. So we have 22*6n/2=66n edges in the converted line graph. **Compared with the original graph with |V| = n and |E| = 6n, the converted line graph with angle information is super large with the node number of 6n and edge number of 66n, which would induce high computational cost.**
>
> - Moreover, different from molecular graphs, **atom number *n* can be super large for crystal graphs.** To be concrete, the maximum number of atoms in a single cell for the Materials Project is 296. Also, we provide the statistics of atom numbers of the two crystal datasets in the following table. Crystal structures have more atoms than regular molecular graphs. Specifically, there are more than 10k samples in The Materials Project with more than 50 atoms in a single cell. Thus, larger complexity will be introduced when adding angular information compared with molecular graphs.
>
> | Dataset | Mean atom numbers in a cell | Max atom numbers in a cell | Number of crystals with > 50 atoms in a cel | Number of crystals with > 100 atoms in a cell |
> |----|----|----|----|----|
> |JARVIS|10.1|140|308|3|
> |MP|29.9|296|11911|1866|
>
> - Beyond this, following your suggestion, we **conducted ablation studies** to show the **high complexity of introducing angular information** to our Matformer in Sec 6.3 and as shown in the table below. We introduced angular information using line graphs following ALIGNN, and added Matformer layers to deal with extra angular information. We use cosine plus RBF (as done by ALIGNN) and SBF kernels to encode angular information. The corresponding running times and final test MAE are shown in the following Table. It can be seen that introducing angular information largely increases the training time by nearly three times, and the performance gain is neglectable. It also shows that our proposed Matformer has great power for periodic graph learning, and is much more efficient.
>
> |JARVIS Formation Energy|MAE|Time Per Epoch|Total Training Time|
> |----|----|----|----|
> |Matformer|0.0325|64 s|8.9 h|
> |Matformer + Angle SBF|0.0332|173 s|23.8 h|
> |Matformer + Angle RBF|0.0325|165 s|22.9 h|
>
> - However, we do believe introducing angular information in periodic graphs properly (without breaking periodic invariance) and efficiently (without introducing large complexity) is challenging yet promising, and we aim to tackle it in future research.
>
> We continue at the 3rd part as below:

---

### Official Review · Reviewer_ifKM · 2022-07-12

**Rating:** 6
**Confidence:** 3
**Soundness:** 3 good
**Presentation:** 2 fair
**Contribution:** 2 fair

**Summary:**

In this paper, the author/s proposed a graph transformer architecture, named Matformer, to infer the physical properties of crystal materials in which their atomic representation is encoded as graphs. The Matformer architecture encodes repeating patterns and is invariant to periodicity; these two properties are fundamental in modeling crystal materials. The performance are assessed on standard benchmarks, considerably outperforming the baselines.

**Questions:**

My questions are reported in the weaknesses

**Limitations:**

I wondered about the model's performance when no periodicity or repetitive patterns in the graph are present.

**Strengths And Weaknesses:**

**Strengths:**

*Originality*
Proposing a model that is invariant to pattern periodicity and that it explicitly captures repetitive patterns in a graph is a novelty.


*Quality*
The quality of this work is good; almost all proofs are present in the paper or in the appendix. The experimental part is clear and the reference adequate.

*Significance*
I think this work is significant for the community since it addresses specific properties of graphs that are not commonly studied.

**Weaknesses:**
* One of your direct competitors is Graphormer, but there are no comparisons, why?
* In line 183 you said "... rigorously prove ..." , where is the proof?
* In line 46 you used $\mathbb{N}$ for the categorical case... the natural number is infinite, do you have infinite classes?
* In Eq 2 the definition of $q_{ij}^h$ consists of the concatenation of $q_i$ three times... is it correct?
* In line 215, you said that one of the differences with respect to Graphormer is that " ...  Graphormer treats every atom as a single node.", but you are doing the same following the definition of $a_i \in \mathbf{A}$. Can you elaborate more?

---

> ### Author Response · Authors · 2022-08-02
> **Graphormer does not consider repeating positions of a given atom in the unit cell; experiments on Matformer without periodicity or repetitive patterns. (2nd Part)**
>
> > 5. In line 215, you said that one of the differences with respect to Graphormer is that " ... Graphormer treats every atom as a single node.", but you are doing the same following the definition of $a_i \in A$. Can you elaborate more?
>
> Thank you for bringing this question up.
>
> - As we state several times in the paper, atom $i$ repeats itself infinitely (like it has infinite duplicates in 3D space) in a crystal structure. The sentence that " ... Graphormer treats every atom as a single node." means Graphormer **does not consider the repeating positions** $p_i + k_1l_1 + k_2l_2 + k_3l_3$ for a given atom $i$ in the unit cell, but **just considers the position $p_i$ of it**. For sure, the same part lies in that, both Graphormer and our method preserve atom features for a given atom, as you said $a_i \in A$ for each atom $i$. However, as we show in the paper, **even though preserving the atom features, ignoring duplicated positions for the same atom will still broke periodic invariance**, as illustrated in Figure 2 also. Hence, in our methods, we treat all the duplicates $p_i + k_1l_1 + k_2l_2 + k_3l_3$ of atom $i$ as the same node in graph construction, which is demonstrated satisfying periodic invariance.
>
> - Secondly, we do an **ablation study to investigate the graph construction method used by Graphormer**, marked as OCgraph in Table 5 in the original paper. For fair comparison, OCgraph is the combination of Graphormer’s graph construction method and Matformer’s architecture. Because of breaking periodic invariance, **the graph construction method used by Graphormer can not produce satisfactory results**. We can see the performance drops from 0.348 to 0.530 when exactly the same model architecture of our Matformer is used.
>
> - Additionally, following your suggestion, we **also conducted a comparison between the original Graphormer and Matformer**, as shown and analyzed in above answer 1 in part 1. The original Graphormer is even much worse than OCgraph.
>
> For the limitations mentioned by reviewer ifKM:
> > 1. I wondered about the model's performance when no periodicity or repetitive patterns in the graph are present.
>
> Thank you very much for letting us know your concerns.
>
> - For this question, as you may have noticed, in the main paper we provided the ablation study of breaking periodic invariance and dropping periodic encodings in Section 6.3.
>
> - For the model's performance without periodicity in the graph, for the same model architecture, using the graph construction method breaking periodic invariance (ignoring periodic information and treating it as a molecular graph) results in a 53% performance drop, and the MAE increased from 0.348 to 0.530 (lower is better).
>
> - For the model's performance without repetitive patterns in the graph, for the same model architecture, dropping periodic patterns encoding leads to a significant performance drop from 0.325 to 0.337.
>
> Hope we have addressed your concerns well, and we are looking forward to your reply. Thanks!

---

> ### Author Response · Authors · 2022-08-02
> **Added Graphormer experiment and comparison; revised paper according to suggestions. (1st Part)**
>
> Thank you for your review and insightful comments. We genuinely appreciate your recognition of the originality, quality, and significance of our paper! For your concerns about the potential weaknesses, we provide a point-wise response as below. We also revised the paper accordingly.
>
> For weaknesses and questions stated above:
> > 1. One of your direct competitors is Graphormer, but there are no comparisons, why?
>
> Thank you for the suggestion.
>
> - Firstly, as mentioned in Section 3.1 (Significance of periodic invariance) in the original version, **Graphormer breaks periodic invariance when applied to periodic graphs**. In Table 5 of the original version & section 6 of the revised version, we **compared two graph construction methods used by Matformer and Graphormer. For fair comparison, the graph construction method used by Graphormer is implemented on top of the Matformer network architecture, denoted as OCgraph in the paper.** Experiments are conducted on JARVIS Formation Energy task to show the influence of breaking periodic invariance. The performance of OCgraph is 0.0530 while our Matformer is 0.0325 (lower is better), which is a significant difference.
>
> - Besides, **following your suggestion, we also conducted experiments compared with the original Graphormer using their official code** on Jarvis dataset Formation Energy task, with some modifications to make it fit in the material data on a single A6000 GPU. The results are shown in the table below. It is worth mentioning that training Graphormer with 1 block and 4 layers on JARVIS formation energy for 500 epochs needs 5 days on one A6000 GPU. The test MAE result for Graphromer is 0.1389, which is worse than all the baseline methods. **We also notice that the training process is unstable for Graphormer, and the training loss is easy to surge. This bad performance and unstable training strongly demonstrate the importance of periodic invariance.** `We can add those results and analysis in the paper if you suggest so.`
>
>      | JARVIS Formation Energy | Test MAE | Total Training Time|
>      |----|----|----|
>      |Matformer| 0.0325| 8.9 h|
>      |Graphormer| 0.1389| 5 days|
>
> > 2. In line 183 you said "... rigorously prove ...", where is the proof?
>
> Thank you for bringing this question up.
>
> For this question, the proof is in Appendix A.2. Thank you for pointing this out and we added a reference in the main paper after that sentence in line 188 to make the proof easier to find.
>
> > 3. In line 46 you used $\mathbb{N}$ for the categorical case... the natural number is infinite, do you have infinite classes?
>
> Thank you for bringing this question up.
>
> We edited it to {$1, 2, \cdots, C$} in the main paper, where $C$ is the number of classes for categorical tasks. We work on the crystal representation learning, and in practice, there usually have finite categories for classification.
>
> > 4. In Eq 2 the definition of $q_{ij}^h$ consists of the concatenation of $q_i$ three times... is it correct?
>
> Yes, your understanding is correct.
>
> - Firstly, as you may notice, there are **three parts of messages** from neighboring node $j$ to the center node $i$; those are, the node feature message from node $i$ and $j$, and the edge feature from edge $e_{ij}$. Hence, we formulate $q_{ij}^h$ to be the concatenation of $q_i$ three times to be in the same dimension with $k_{ij}^h$, which is the concatenation of $k_i, k_j$ and $e_{ij}^{h’}$. Importantly, by doing this, we let $q_i$ attend each of the three components in $k_{ij}^h$. As a result, we calculate the attention coefficients for the whole message containing adequate information from node $i$, $j$ and edge $e_{ij}$. This is because we compute a similarity score vector for each of $(q_i, k_i), (q_i, k_j)$, and ($q_i, e_{ij})$, as in Eq $2$. **We revised the texts following in Eq $2$ to clarify this.**
>
> - Besides, we also provided a detailed illustration of operations in Matformer Layer in Figure 2 of Appendix, which you may have already noticed. If any confusion still exists, please let us know. We are more than glad to make it clearer.
>
> We continue at the 2nd part as below:

---

> ### Author Response · Authors · 2022-08-08
> **Dear Reviewer ifKM**
>
> Thank you very much for the insightful questions and your precious time!
>
> We conducted comparison experiments with Graphormer and revised the paper according to your valuable suggestions. We also provided clarifications and answers for your questions.
>
> As the deadline for discussion periodic is approaching, we are looking forward to your reply and are more than glad to answer any other questions.

---

> ### Author Response · Authors · 2022-08-09
> **Author's follow-up to reviewer ifKM, half-day before author-reviewer conversation ends**
>
> Dear Reviewer ifKM,
>
> Thanks again for your valuable comments and suggestions in your initial review, which helps improve our work a lot. Regarding your main concerns on **comparison with Graphormer**, **clarifications in writing**, and **performances when no periodicity or repetitive patterns in the graph**, we have conducted substantial experiments and also revised the paper heavily in our rebuttal on August 1st. Could you please check at your earliest convenience? Thank you so much!
>
> **For comparison with Graphormer**, **we followed your suggestion and conducted experiments comparing with the original Graphormer using their official code.** From the comparison, we notice that the powerful Graphormer can only achieve 0.1389 in terms of test MAE, which is worse than all baseline methods. Performance of our Matformer is 0.0325. We also notice that the training process is unstable for Graphormer. Graphormer fails to integrate periodic invariance. The bad performance and unstable training strongly demonstrate the importance of periodic invariance.
>
> **For some clarifications**, we have **revised our paper heavily to make it clearer to readers.**
>
> Regarding **performances when no periodicity or repetitive patterns in the graph**, we conducted ablation studies in our original paper and **provided further clarifications to make it clearer.**
>
> **We hope that you could reply to our rebuttal and consider raising your score if we have addressed your concerns. Also, please let us know if there are any additional concerns or feedback. Thank you!**
>
> Sincerely,
> Authors

---

### Official Review · Reviewer_VsGM · 2022-07-18

**Rating:** 6
**Confidence:** 4
**Soundness:** 3 good
**Presentation:** 4 excellent
**Contribution:** 3 good

**Summary:**

 The paper studies representation learning for periodic objects such as crystal. The paper formally defines the periodic invariance property and periodic pattern encoding. To achieve both periodic invariance and periodic pattern encoding, the author proposes two type of constructions of local graphs, modified from existing methods (CGCNN and Graphnormer). The author further develops a proper message passing based transformer over the proposed graph construction method. Overall the paper is well-written and informative for representation learning on periodic objects. The experimental result shows the effectiveness of their designs.

**Questions:**

Please see previous comments.

**Strengths And Weaknesses:**

Strengths:
1. The written is extremely clear and informative in terms of analysis and literature review. Reading the paper is a kind of joy. The author also clearly state the difference with respect to several similar previous works.
2. The proposed method achieves all invariant properties desired. The design is simple, clean, and meaningful.
3. The proposed properties for periodic objects are formal and informative. if these properties are proposed the first time here, this is a good contribution.

Weaknesses:
1. Even stated several times, the proposed graph construction methods are similar to previous works. In this perspective, the novelty is a bit reduced.
2. The performance improvement comparing with the strong baseline ALIGNN is small. Given ALIGNN uses angular information, it would interesting if the author can adapt angular information to the designed Matformer. If the author can further improve the current Matformer, this paper will become extremely good.

---

> ### Author Response · Authors · 2022-08-02
> **Significant performance gains beyond ALIGNN in terms of EwT and wT & overall conclusion for Question 2 (3rd Part)**
>
> `Thirdly, beyond this,` to show the significant performance gains of our Matformer beyond ALIGNN, we also **added two new metrics**, including energy within threshold (EwT) and prediction within threshold (wT) for evaluation. These metrics are from paper [1], and are **well recognized by the community. This is because they provide a different perspective from MAE (or RMSE), and are meaningful in practice**. For example, EwT measures the percentage of estimated energies that are likely to be practically useful when the absolute error is within a certain threshold, and this threshold could be 0.01 or 0.02.
> **We then conducted comparison with ALIGNN in terms of EwT and wT to better demonstrate the performance gains.**
>
> - **For all the tasks in JARVIS and formation energy and bandgap tasks in the Materials Project, we use the official prediction results for test sets from ALIGNN to calculate corresponding results. The bulk moduli and shear moduli are not included because no official results from ALIGNN are released.** We use EwT 0.02 to denote energy prediction error within the threshold of 0.02 and  EwT 0.01 to denote energy prediction error within the threshold of 0.01 for tasks of formation energy and total energy. We use wT 0.02 to denote prediction error within the threshold of 0.02 and wT 0.01 to denote prediction error within the threshold of 0.01 for other tasks. Results are reported below.
>
>      | Formation Energy JARVIS | EwT 0.01 | EwT 0.02 | Formation Energy MP | EwT 0.01 | EwT 0.02 |
>      |----|----|----|----|----|----|
>      |ALIGNN|39.59%|59.64%||49.94%|71.10%|
>      |Matformer|41.17%|60.25%||55.86%|75.02%|
>
> - It can be seen from this table that Matformer has more accurate predictions for crystals. Interestingly, the performance gains of our Matformer beyond ALIGNN in terms of EwT mainly **come from more accurate energy predictions within an absolute error of 0.01.** This indicates that Matformer generates more accurate predictions when the prefixed error (threshold) is more strict. Compared with JARVIS, the Materials Project has 15422 more training samples. With more training samples available, the percentage of predicted energies obtained by Matformer within 0.01 increases by 14.69%, which is significantly better than ALIGNN, revealing **the huge potential of Matformer when larger crystal dataset is available.** We also revised Section 6.2 of the paper and added experiments and discussions on the new metric EwT.
>
>      | Band Gap (MBJ) JARVIS  | wT 0.01 | wT 0.02 | Band Gap (OPT) JARVIS  | wT 0.01 | wT 0.02 |
>      |----|----|----|----|----|----|
>      |ALIGNN|18.10%|31.87%||48.73%|61.79%|
>      |Matformer|31.70%|43.64%||56.31%|64.02%|
>
> - It can be seen from the table that there are 13.6% gain in terms of wT 0.01 for Band Gap (MBJ) and 7.58% gain in terms of wT 0.01 for Band Gap (OPT), which are very significant.
>
>
>      | Ehull JARVIS | wT 0.01 | wT 0.02 | Total Energy JARVIS | EwT 0.01 | EwT 0.02 |
>      |----|----|----|----|----|----|
>      |ALIGNN|22.92%|39.52%||35.09%|55.20%|
>      |Matformer|28.37%|44.81%||36.84%|57.36%|
>
>
>      | MP-BandGap | wT 0.01 | wT 0.02 |
>      |----|----|----|
>      |ALIGNN|19.79%|30.15%|
>      |Matformer|27.01%|35.22%|
>
> Overall, **Matformer outperforms ALIGNN in terms of MAE, EwT 0.01, EwT 0.02, wT 0.01, and wT 0.02 for all these seven tasks in JARVIS and the Materials Project, demonstrating that our Matformer is more powerful (in different metrics thus from different perspectives) than previous state-of-the-art ALIGNN.** We included the comparison on formation energy-MP, formation energy-JARVIS, and total energy-JARVIS in Section 6.2 of the revised paper due to space limit. **All the results can be added to the main paper if you suggest so.** Additionally, **Matformer is three times faster than ALIGNN and nearly 3 times faster than Matformer + Angle.** When angular information is added to our Matformer, not much performance gain is shown. We think **periodic invariant graph construction and periodic patterns encoding in Matformer already contain sufficient information to distinguish different crystals.**
>
> Hope we have addressed your concerns well, and we are looking forward to your reply. Thanks!
>
> > Reference
>
> [1] Chanussot, Lowik, et al. "Open catalyst 2020 (OC20) dataset and community challenges." ACS Catalysis 11.10 (2021): 6059-6072.

---

> > ### Comment · Reviewer_VsGM · 2022-08-08
> > **Thank you for detailed response**
> >
> > Thank very much for adding additional explanation and result. I gave score 6 but in my mind it was between 6 and 5, now I'm more certain this paper worth 6 score.

---

> > > ### Author Response · Authors · 2022-08-08
> > > **Dear Reviewer VsGM**
> > >
> > > Thank you very much for your precious time and your recognition of our work! Your valuable and insightful comments and suggestions help improve our work a lot.
> > >
> > >
> > > Sincerely,
> > >
> > > Authors

---

> ### Author Response · Authors · 2022-08-02
> **Consistently better results than ALIGNN in MAE ; revised paper: added complexity analysis and experiments for combining Matformer with angles (2nd Part)**
>
> > 2. The performance improvement compared with the strong baseline ALIGNN is small. Given ALIGNN uses angular information, it would be interesting if the author can adapt angular information to the designed Matformer. If the author can further improve the current Matformer, this paper will become extremely good.
>
> Thank you for your recognition.
>
> `Firstly,` about the performance gains beyond the strong baseline ALIGNN, **our Matformer consistently outperforms ALIGNN on two widely used material datasets for 9 different tasks in terms of MAE**, and is **three times faster** in total training time and nearly three times faster in reference time.
>
> `Secondly,` we want to particularly point out that, we aim to **achieve both effectiveness and efficiency**. Without using angles, our Matformer is three times faster than ALIGNN in training time, and better than ALIGNN in performance. **We further provided more analysis and experimental results as below and in Section 6.3 and Appendix A.6 in our revised paper.**
>
> - Firstly, **adapting angular information introduces large complexity.** Assume that we have $n$ atoms in a single cell and thus $n$ nodes in the original graph. Also assume that every node has at least 12 neighbors (following the graph construction method of Matformer) and there are no self-connecting edges. This will result in a graph $G=(V,E)$ where $|V| = n$ and $|E| = 12/2 n = 6n$. When converting graph $G$ into $L(G)$, every edge is treated as a node in the line graph. So we have $6n$ nodes in the line graph. Every edge in the original graph is connecting $2$ nodes and every node has $(12 - 1)$ other edges, resulting in $22$ neighboring edges for each edge. So we have $22*6n/2=66n$ edges in the converted line graph. Compared with the original graph with $|V| = n$ and $|E| = 6n$, ** the converted line graph with angles is super large with the node number of $6n$ and edge number of $66n$, which would induce high computational cost.**
>
> - Also, following your suggestion, we **conducted experiments to adapt angular information in Matformer .** We add two edge layers in the original Matformer to deal with extra angular information. We use cosine plus RBF (as done by ALIGNN) and SBF kernels to encode the angular information. The corresponding running times and final test MAE are shown in the following Table.
>
>
>      |JARVIS Formation Energy|MAE|Time Per Epoch|Total Training Time|
>      |----|----|----|----|
>      |Matformer|0.0325|64 s|8.9 h|
>      |Matformer + Angle SBF|0.0332|173 s|23.8 h|
>      |Matformer + Angle RBF|0.0325|165 s|22.9 h|
>
> - It can be seen from this table that introducing angular information by using line graphs will result in **nearly 3 times training time per epoch and in total, without much performance gain.** The main reason that ALIGNN (also the famous model DimeNet [1] for molecules) uses angles lies in that, angle information helps determine the shape of a geometric graph thus improving the model performance. We think periodic invariant graph construction add periodic patterns encoding in Matformer *already contain sufficient information to distinguish different crystals.* We revised the paper and added those results along with analysis in ablation study in Section 6.3 and Appendix A.6.
>
> - We do believe it is an excellent point to improve the current Matformer by introducing angular information properly to satisfy both periodic invariance and to encode periodic patterns with relatively low time complexity. The research on periodic graphs is new but promising, and there still exist a lot of ideas worth exploring. We aim to tackle them (especially integrating the angle information effectively and efficiently) in future research.
>
> > Reference
>
> [1] Klicpera, Johannes, Janek Groß, and Stephan Günnemann. "Directional message passing for molecular graphs." ICLR 2020.
>
> We continue at the 3rd part as below:

---

> ### Author Response · Authors · 2022-08-02
> **Novelty: discover and formally define two important components for crystal graph construction; propose two solutions to preserve periodic invariance and a new, elegant, and efficient way to encode periodic patterns; Matformer is powerful and efficient (1st Part)**
>
> We genuinely appreciate your review and insightful comments. Below we provide a point-wise response to address the weaknesses. We also revised the paper accordingly.
>
> For weaknesses and questions stated above:
> > 1. Even stated several times, the proposed graph construction methods are similar to previous works. In this perspective, the novelty is a bit reduced.
>
> We kindly disagree with this point.
>
> - Firstly, in this work, we **discover and formally define two important components for crystal graph construction**: periodic invariance and periodic patterns encoding. Firstly, **periodic invariance is of great importance for periodic structures like crystals, but is rarely noticed by previous works.** Treating atoms as individual nodes (like in molecular graphs) breaks periodic invariance. However, this strategy is widely used when handling periodic structures, even in some powerful methods such as Graphormer. **Breaking periodic invariance will result in different representations for the same crystal structure, which could confuse machine learning models and produce suboptimal performance.** Besides, when dealing with periodic crystals, no previous work tried to encode periodic patterns. Without encoding periodic patterns, the graph representation only captures a local unit (local structure), but fails to capture how the unit (local structure) expands itself in 3D space. **Such an important component is even not noticed, not to mention solved in existing works.** To this end, we believe the discovery and formal definition of periodic invariance and periodic patterns encoding are of great value, importance, and significance to the community.
>
> - More importantly, we also **propose two solutions to preserve periodic invariance**, and design **a new, elegant, and efficient way to encode periodic patterns** when constructing graphs. *Also, the proposed Matformer is demonstrated to be* **powerful** (outperforms previous SOTA consistently) and **efficient** (nearly 3 times faster in training and inference time). Due to the nature of period graphs, Matformer is very different from transformers for texts, images, and regular graphs. All the proposed methods in this work are new, effective, and efficient.
>
> - Overall, we think the novelty is sufficient due to 1). we discover, formally define, and propose approaches to resolve two vital components for crystal graph construction; and 2). we provide a novel, powerful, and efficient crystal learning model.
>
> We continue at the 2nd part as below:

---

### Meta-Review · Area_Chair_TTQD · 2022-08-26

**Recommendation:** Accept
**Confidence:** Less certain

**Metareview:**

This paper had borderline reviews. While the reviewers felt that the method was novel and the presentation very good, they also cited weaknesses such as limited performance gains over baselines and limited novelty. The authors responded in detail to many of the concerns, including with additional experiments, causing several reviewers to increase their scores. We encourage the authors to incorporate aspects of these responses in the final version.


Note: As the AC I will note that I did not find the authors’ very long summary of the discussion helpful in making my decision. I briefly skimmed but did not fully read it. Such a summary/discussion from the authors will obviously be presented through a very biased lens, and so it is much more helpful for me as an AC to directly look at the discussion myself in making a decision, and to ask follow up questions to directly to the reviewers if any clarification is needed.

**Award:**

No

---

### Decision · Program_Chairs · 2022-09-14

Accept